# NEURAL GAUSSIAN RADIO FIELDS FOR CHANNEL ESTIMATION

## ABSTRACT

Accurate channel state information (CSI) is a significant bottleneck in modern wireless networks, with pilot overhead consuming 11-21% of transmission bandwidth and feedback delays causing severe throughput degradation under mobility. To address this, this work introduces a new class of neural fields designed for coherent wave-based phenomena, called *neural Gaussian radio fields* (nGRF), which combines the efficiency of explicit primitive-based representations with a novel differentiable operator. nGRF replaces view-dependent, computer graphics-centric rasterization with direct, complex-valued aggregation in 3D space that natively models wave superposition and interference. Consequently, this reframes the learning objective from a function-fitting task to a well-posed source-recovery problem. In evaluations, nGRF demonstrates superior performance across diverse environments. In indoor scenarios, it achieves 10.9 dB higher prediction SNR than state-of-the-art methods while reducing inference latency from 242 ms to 1.1 ms (a $220\times$ speedup). For large-scale outdoor environments where existing approaches fail, nGRF achieves an SNR of 26.2 dB. Furthermore, the proposed method reduces the required measurement density by $18\times$ (0.011 vs. 0.2-178.1 measurements/ft$^3$) and cuts the training time from hours to minutes (a $180\times$ reduction), enabling rapid adaptation to dynamic environments. The code and datasets are available at `https://github.com/anonym-auth/n-grf`.

## 1 INTRODUCTION

Despite decades of work, real-time channel state information (CSI) *estimation and prediction* remains the principal unresolved bottleneck in both current and next-generation wireless networks. CSI, represented by the complex matrix $\mathbf{H}$, captures how signals propagate through direct paths, reflections, diffractions, and scattering between the transmitter and receiver. Accurate CSI enables transmitters to adapt their waveforms, power levels, and spatial precoding to channel conditions, directly determining achievable data rates and link reliability Bai & Atiquzzaman (2003). The sub-millisecond latencies and Gbps data rates targeted by 5G and 6G networks demand high-fidelity, low-overhead CSI estimation Giordani et al. (2020).

Multiple-input multiple-output (MIMO) technology uses multiple antennas at the transmitter and receiver to send multiple parallel data streams over the same frequency band. This *spatial multiplexing* boosts data rates and capacity without extra bandwidth. With $N$ antennas at each end, the theoretical peak throughput scales linearly by a factor of $N$. Massive MIMO further increases the antenna arrays by factors of ten, making $\mathbf{H}$ difficult to characterize Balevi et al. (2020). Two fundamental challenges make CSI estimation and prediction difficult to resolve. Firstly, pilot overhead is a significant bottleneck. Because every resource element dedicated to pilots cannot be used for data, this overhead directly diminishes the achievable throughput. In current 5G NR systems, pilots consume as much as 11% to 21% of the transmission bandwidth, a proportion that is derived from the structure of the physical resource block (PRB) Dahlman et al. (2014); Lin (2022). This issue is exacerbated in high-mobility environments where channels change rapidly, and in cell-free massive MIMO systems where feedback overhead can reach 57 Gb/s Jardosh et al. (2005). Thus, a trade-off arises: reducing pilots is essential for efficiency, but doing so risks inaccurate CSI, which also degrades system performance.

Secondly, even after channel information is acquired, it quickly becomes outdated in *dynamic environments* due to channel aging Truong & Heath (2013). Under mobility, wireless channels decorrelate in milliseconds, so any delay between CSI measurement and its usage can render it obsolete. For example, at a mid-band frequency of 3.5 GHz, a user moving at urban driving speeds ( 30 km/h) experiences a channel coherence time of 2 ms Wang et al. (2024). A 4 ms feedback delay under these conditions can cut the data rate by approximately 50%. A mmWave channel can decorrelate within a single 1-ms 5G subframe. Furthermore, dense networks face *pilot contamination*, where inter-cell pilot reuse degrades CSI accuracy, a known bottleneck for MIMO systems Elijah et al. (2015).

Many AI-driven CSI estimators, however, disregard the physical 3D structure governing radio propagation. Data-driven methods, such as generative models or recurrent networks, treat the channel as an abstract data vector, lacking the inductive bias of the underlying physics and often incurring high latency from iterative sampling or large network backbones Arvinte & Tamir (2022); Aldossari & Chen (2019). Recent neural field approaches, while spatially aware, suffer from their own architectural limitations. Neural radiance fields (NeRF)-based models rely on slow, implicit representations that require computationally expensive volumetric integration for every channel query Lu et al. (2024); Zhao et al. (2023). Concurrently, methods adapting 3D Gaussian splatting (3DGS) for channel modeling are ill-suited for the task; they regress scalar power and apply 2D projections made for visual rendering, failing to capture the complex-valued nature of electromagnetic fields Wen et al. (2025); Niemeyer et al. (2025). These limitations motivate a new modeling design that is both computationally efficient and physically grounded.

This work introduces *neural Gaussian radio fields (nGRF)*, a new class of neural fields designed for coherent wave-based phenomena that combines the efficiency of explicit primitives with a novel differentiable operator. The framework replaces view-dependent, computer graphics-centric rasterization with direct, complex-valued aggregation in 3D space that models wave superposition. Each 3D Gaussian acts as a localized *radio modulator*, whose properties are learned conditioned on the transmitter's position. The channel at any receiver location is then rendered as a spatially weighted superposition of these contributions. This approach reframes the learning objective from a function-fitting task to a source-recovery problem, resulting in a model that is faster and more accurate than prior art.

**Contributions.** This work introduces nGRF, a framework that uses explicit 3D Gaussian primitives for MIMO channel matrix estimation by modeling electromagnetic field propagation through direct 3D aggregation. nGRF demonstrates that structured, explicit representations supplemented by physical principles can provide a better and more efficient way to model complex field phenomena than generic deep learning architectures. Specifically, it achieves, on average, a 10.9 dB higher SNR than state-of-the-art methods across diverse environments while supporting both SISO and MIMO configurations. For practical deployment in 5G NR systems, nGRF reduces the pilot overhead from thousands of resource elements to just 96 bits of position data, decreasing resource grid occupation from 11-21% to 0.2%. In terms of computational efficiency, evaluations show that nGRF requires $18\times$ lower measurement density, trains $180\times$ faster, and offers $220\times$ lower inference latency compared to NeRF-based alternatives, thus enabling rapid adaptation to new environments from sparse measurements within minutes. Moreover, despite being trained on a single carrier frequency, nGRF generalizes across frequencies and predicts the channel response across a wide band of subcarriers. This strongly suggests that the model learns the underlying, frequency-agnostic principles of wave propagation within the environment rather than memorizing frequency-specific mappings.

## 2 RELATED WORK

Prior work on CSI estimation spans three main approaches: data-driven models that disregard spatial physics, implicit neural fields that are computationally prohibitive, and explicit 3DGS-based methods that misapply visual rendering techniques to wave phenomena. The limitations of each motivate the need for a new approach.

**Data-Driven Approaches.** Early deep learning models treated CSI estimation as a data-driven regression or generation task. Regressors using LSTMs or DNNs map pilot measurements to channel estimates with low latency, but offer limited accuracy and struggle to generalize Aldossari & Chen (2019). Generative models, including score-based diffusion Arvinte & Tamir (2022); Jin et al. (2024)

and variational autoencoders Chen et al. (2025), learn rich channel priors to achieve high accuracy. However, their reliance on iterative sampling or large U-Net backbones results in high inference latency ($\geq$100 ms) and significant memory requirements, rendering them impractical for real-time applications. These models fundamentally treat the channel as an abstract vector and miss the inductive biases provided by the spatial structure of the radio propagation environment.

**Implicit Neural Fields.** To incorporate spatial context, recent work has adapted NeRF to radio frequency modeling. Methods like NeRF$^2$ Zhao et al. (2023) and NeWRF Lu et al. (2024) represent the radio environment as a continuous, implicit function $F_\Theta$ learned by a large MLP. While these models capture the 3D nature of the scene, their implicit representation is their primary bottleneck. Estimating the channel $\mathbf{H}$ requires numerically approximating the volumetric rendering integral along a ray $\mathbf{r}(t)$ as

$$\mathbf{H}(\mathbf{r}) = \int_{t_{near}}^{t_{far}} T(t)\sigma(\mathbf{r}(t))\mathbf{c}(\mathbf{r}(t))dt, \text{ where } (\sigma, \mathbf{c}) = F_\Theta(\mathbf{r}(t), \mathbf{d}), \tag{1}$$

where $\mathbf{r}(t)$ is a ray with direction $\mathbf{d}$, $\sigma(\cdot)$ is the learned volume density representing energy attenuation, $\mathbf{c}(\cdot)$ is the complex-valued field contribution at a point, and $T(t) = \exp(-\int_{t_{near}}^{t} \sigma(\mathbf{r}(s))ds)$ is the accumulated transmittance along the ray. This integral is discretized into a sum over $N_s$ points, requiring $N_s$ separate evaluations of the MLP $F_\Theta$ for every ray. The resulting computational complexity leads to high inference latency (200-350 ms), making these methods too slow for the millisecond-scale requirements of modern wireless systems.

**3DGS-based Methods.** To address the speed limitations of NeRFs, another line of work has explored explicit representations based on 3DGS. Approaches like WRF-GS Wen et al. (2025) model the environment as a collection of 3D Gaussians, enabling near real-time rendering. However, these methods are constrained by their origin in visual rendering. They project 3D primitives onto a 2D plane and synthesize a final value using alpha compositing, where the final color $C$ is computed as

$$C = \sum_{i=1}^{N} c_i \alpha_i \prod_{j=1}^{i-1} (1 - \alpha_j), \tag{2}$$

where the final value $C$ is a sum over $N$ Gaussians sorted by depth, each having a color (or feature) $c_i$ and an opacity $\alpha_i$. This formulation models occlusion, where foreground objects block background ones. This is physically incorrect for electromagnetic waves, which do not occlude but rather superimpose. The total field at a receiver should be the complex-valued summation of all contributing fields, a principle that alpha blending fails to capture. Consequently, these methods have been limited to regressing scalar quantities such as signal strength (RSSI) rather than the complete CSI needed for optimal MIMO performance.

The limitations of the prior art reveal the need for a new model that combines the strengths of existing approaches while avoiding their architectural flaws. An ideal solution must utilize an explicit representation for speed, but it must also be designed from the ground up for electromagnetic physics. nGRF is introduced as a direct response to this need.

## 3 NEURAL GAUSSIAN RADIO FIELDS

### 3.1 PROBLEM DESCRIPTION

Channel estimation fundamentally requires solving Maxwell's equations for a given environment. For time-harmonic electromagnetic fields at frequency $\omega$, this reduces to solving the vector Helmholtz equation expressed as $\nabla \times \nabla \times \mathbf{E}(\mathbf{r}) - k^2\mathbf{E}(\mathbf{r}) = -j\omega\mu_0\mathbf{J}(\mathbf{r})$, where $\mathbf{E}(\mathbf{r})$ is the electric field at position $\mathbf{r} \in \mathbb{R}^3$, $k = \omega\sqrt{\mu\epsilon}$ is the wavenumber, and $\mathbf{J}(\mathbf{r})$ represents the current sources. Solving this partial differential equation (PDE) with the appropriate boundary conditions defined by the environment's geometry and materials would yield perfect CSI. However, this is computationally intractable for any non-trivial scene.

The solution to the Helmholtz equation can be expressed via a Green's function $G(\mathbf{r}, \mathbf{r}')$, which describes the field at $\mathbf{r}$ due to a point source at $\mathbf{r}'$. In free space, this is a spherical wave, $G_0(\mathbf{r}, \mathbf{r}') = e^{ik|\mathbf{r}-\mathbf{r}'|}/(4\pi|\mathbf{r} - \mathbf{r}'|)$, while in a complex environment, the total field is the sum of the incident field

from the source and the scattered field from all interacting objects. The scattered field, in turn, can be described by an integral of the Green's function over the surfaces of all scatterers. Thus, the complex multipath channel results from the superposition of waves originating from a set of effective sources distributed throughout the environment.

This motivates representing the field using a basis of functions that can model these localized wave contributions. Under the paraxial approximation, where waves propagate primarily along a single direction, solutions to the Helmholtz equation take the form of Gaussian beamsLiu et al. (2014). Such a physical connection suggests that a superposition of anisotropic 3D Gaussian functions can serve as a flexible basis set for representing the electromagnetic field in its entirety.

Existing methods make different trade-offs. Ray tracing, for example, approximates the solution in the geometric optics limit (wavelength $\lambda \to 0$), treating waves as simple rays. This fails to capture wave phenomena like diffraction and sub-wavelength interference, which are needed for accurate channel modeling. Implicit neural fields based on NeRF Mildenhall et al. (2020) attempt to learn a continuous volumetric representation of the field. However, these models lack physical priors for wave propagation. They are generic function approximators that must learn the field's structure from scratch, requiring dense measurements and computationally expensive volumetric integration to render a single channel estimate.

nGRF is founded on the principle that while the electromagnetic field is continuous, the environmental features that cause complex multipath propagation (scatterers, reflectors, diffractive edges) are spatially localized. The total field $\mathbf{E}(\mathbf{r})$ can be decomposed into a line-of-sight component and a scattered field $\mathbf{E}_{\text{scat}}(\mathbf{r})$, which arises from these interactions. This scattered field can be modeled as a superposition of contributions from a discrete set of effective scattering centers. Mathematically, this is expressed as

$$\mathbf{E}(\mathbf{r}) \approx \mathbf{E}_{\text{LoS}}(\mathbf{r}) + \sum_{i=1}^{N} \mathbf{E}_i(\mathbf{r}) = \mathbf{E}_{\text{LoS}}(\mathbf{r}) + \sum_{i=1}^{N} \mathcal{A}_i(\mathbf{p}_{\text{tx}}, \mathbf{p}_{\text{rx}}) G_i(\mathbf{r}; \boldsymbol{\mu}_i, \boldsymbol{\Sigma}_i), \tag{3}$$

where each $\mathbf{E}_i(\mathbf{r})$ represents the field contribution from the $i$-th scattering region, modeled by a 3D Gaussian primitive $G_i$ with mean $\boldsymbol{\mu}_i$ and covariance $\boldsymbol{\Sigma}_i$, and where the complex amplitude $\mathcal{A}_i$ depends on the positions of the transmitter and receiver. Formulating the problem in this way transforms the intractable task of solving a PDE into a well-posed source-recovery problem. The objective is no longer to fit a function to data, but rather to infer the parameters of a set of adaptive basis functions, where each function represents an effective scattering source that contributes to the total field.

## 3.2 nGRF Design

nGRF models the radio environment as a set of $N$ physics-constrained 3D Gaussian primitives $\{G_i\}_{i=1}^{N}$. Each primitive is a localized basis function, or "radio modulator," that represents a component of the total electromagnetic field. The overall architecture is illustrated in Figure 1.

Each Gaussian primitive is parameterized by its geometric properties and learned electromagnetic attributes. The geometric properties include its mean position $\boldsymbol{\mu}_i \in \mathbb{R}^3$ and its covariance matrix $\boldsymbol{\Sigma}_i \in \mathbb{R}^{3 \times 3}$, which defines its shape and orientation. The unnormalized density of the $i$-th Gaussian at a point $\mathbf{x} \in \mathbb{R}^3$ is given by

$$G_i(\mathbf{x}) = \exp\left(-\frac{1}{2}(\mathbf{x} - \boldsymbol{\mu}_i)^T \boldsymbol{\Sigma}_i^{-1}(\mathbf{x} - \boldsymbol{\mu}_i)\right). \tag{4}$$

To ensure the covariance matrix $\boldsymbol{\Sigma}_i$ remains positive semi-definite during optimization, it is parameterized by a rotation matrix $\mathbf{R}_i \in \text{SO}(3)$ (special orthogonal group) and a diagonal scaling matrix $\mathbf{S}_i = \text{diag}(s_{i,1}, s_{i,2}, s_{i,3})$, such that $\boldsymbol{\Sigma}_i = \mathbf{R}_i \mathbf{S}_i \mathbf{S}_i^T \mathbf{R}_i^T$. The rotation is further represented by a unit quaternion $\mathbf{q}_i \in \mathbb{R}^4$, and the scaling factors are parameterized via an exponential map, $s_{i,j} = \exp(s'_{i,j})$, to ensure positivity.

The electromagnetic attributes of each Gaussian are determined by a pair of neural networks, as shown in Figure 2. These networks learn the complex relationship between spatial locations and the resulting field contributions. Specifically, the `AttributeNetwork`, $f_{\text{attr}}$, conditions the properties of each Gaussian on the transmitter's location. It maps the Gaussian's mean $\boldsymbol{\mu}_i$ and the

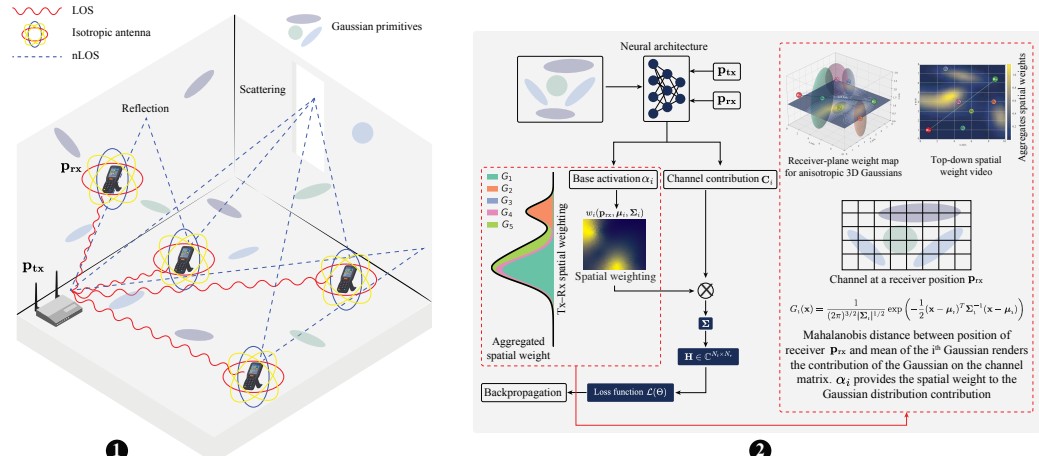

Figure 1: **An illustration of the nGRF framework.** The framework represents the wireless environment using a collection of 3D Gaussian primitives, each encoding localized electromagnetic field contributions which are learned via neural networks and aggregated to synthesize the final channel matrix.

transmitter's position $\mathbf{p}_{\text{tx}}$ to a latent feature vector $\mathbf{z}_i \in \mathbb{R}^d$ and a scalar base activation $\alpha_i \in \mathbb{R}$ as $(\mathbf{z}_i, \alpha_i) = f_{\text{attr}}(\gamma_L(\boldsymbol{\mu}_i), \gamma_L(\mathbf{p}_{\text{tx}}); \Theta_{\text{attr}})$. To capture the high-frequency spatial variations present in wave phenomena (with wavelength $\lambda$), the input coordinates are first transformed using a multi-resolution positional encoding $\gamma_L(\cdot)$ Mildenhall et al. (2020) defined as

$$\gamma_L(\mathbf{x}) = \left[ \mathbf{x}, \sin(2^0\pi\mathbf{x}), \cos(2^0\pi\mathbf{x}), \dots, \sin(2^{L-1}\pi\mathbf{x}), \cos(2^{L-1}\pi\mathbf{x}) \right]. \quad (5)$$

Positional encoding allows the network to effectively learn functions with fine-grained detail, which is necessary to model phase changes that occur on the scale of a wavelength.

Furthermore, the latent vector $\mathbf{z}_i$ encodes the complex electromagnetic behavior of the scattering region, while the activation $\alpha_i$ controls its overall contribution strength. The `DecoderNetwork`, $f_{\text{dec}}$, then maps this latent vector to the final complex-valued channel contribution matrix $\mathbf{C}_i \in \mathbb{C}^{N_t \times N_r}$ for a MIMO system with $N_t$ transmit and $N_r$ receive antennas, given by $\mathbf{C}_i = f_{\text{dec}}(\mathbf{z}_i; \Theta_{\text{dec}})$. Since neural networks output real values, $\mathbf{C}_i$ is split into its real and imaginary parts, $\mathbf{C}_i = \mathbf{C}_i^{\text{re}} + j\mathbf{C}_i^{\text{im}}$. Together, these components allow each Gaussian primitive to function as a fully learned, localized radio modulator.

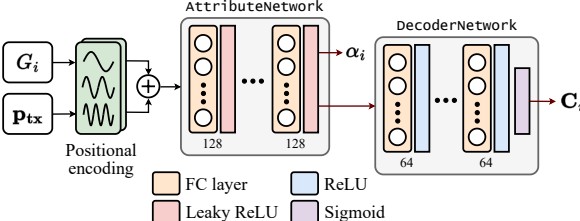

Figure 2: **Neural architecture.** The *AttributeNetwork* processes Gaussian and transmitter positions through separate positional encoders to produce latent features and base activation logits. The *DecoderNetwork* then transforms the latent features from *AttributeNetwork* into complex channel contributions.

### 3.3 CHANNEL RENDERING

In nGRF, channel rendering computes the MIMO channel matrix $\mathbf{H}$ at a specific receiver location $\mathbf{p}_{\text{rx}}$ through a direct, weighted aggregation of the contributions from all Gaussian primitives. This process is a learned, discrete implementation of the superposition principle that governs linear wave phenomena, and eliminates the need for computationally expensive ray marching or volumetric integration. For a given transmitter-receiver pair $(\mathbf{p}_{\text{tx}}, \mathbf{p}_{\text{rx}})$, the channel matrix $\mathbf{H}(\mathbf{p}_{\text{rx}}, \mathbf{p}_{\text{tx}}) \in \mathbb{C}^{N_t \times N_r}$ is computed as $\mathbf{H}(\mathbf{p}_{\text{rx}}, \mathbf{p}_{\text{tx}}) = \sum_{i=1}^{N} w_i(\mathbf{p}_{\text{rx}}, \boldsymbol{\mu}_i, \boldsymbol{\Sigma}_i) \cdot \mathbf{C}_i$, where $\mathbf{C}_i$ is the complex contribution from the $i$-th Gaussian (output by the decoder network), and $w_i$ is a spatial weighting function that

determines its influence at the receiver's location. This weight is formally defined as

$$w_i(\mathbf{p}_{\text{rx}}, \boldsymbol{\mu}_i, \boldsymbol{\Sigma}_i) = \alpha_i \cdot \exp\left(-\frac{1}{2}(\mathbf{p}_{\text{rx}} - \boldsymbol{\mu}_i)^T \boldsymbol{\Sigma}_i^{-1}(\mathbf{p}_{\text{rx}} - \boldsymbol{\mu}_i)\right), \tag{6}$$

where $\alpha_i$ is the base activation and the exponential term is derived from the Gaussian density function. This formulation has a clear physical interpretation: the influence of each primitive decays according to the Mahalanobis distance between its center and the receiver. The anisotropic covariance $\boldsymbol{\Sigma}_i$ allows this influence to be directional and effectively models phenomena like specular reflections from planar surfaces. Directly aggregating in 3D space this way is fundamentally different from the 2D projection and alpha-compositing used in 3DGS-based methods. Mainly, it correctly models the additive nature of electromagnetic fields rather than visual occlusion, making it well-suited for radio propagation.

### 3.4 CONVERGENCE & OPTIMIZATION

The model parameters $\Theta = \{\boldsymbol{\mu}_i, \mathbf{q}_i, \mathbf{s}'_i\}_{i=1}^N \cup \{\Theta_{\text{attr}}, \Theta_{\text{decoder}}\}$ are optimized by minimizing a composite loss function $\mathcal{L}(\Theta)$ using stochastic gradient descent. The loss function combines a primary estimation error term with two regularization terms and is defined as $\mathcal{L}(\Theta) = \mathcal{L}_{\text{est}} + \lambda_{\text{act}}\mathcal{L}_{\text{activation}} + \lambda_{\text{reg}}\mathcal{L}_{\text{regularization}}$.

The estimation loss $\mathcal{L}_{\text{est}}$ measures the discrepancy between the predicted and ground-truth channel matrices using the Frobenius norm, which is sensitive to errors in both amplitude and phase, and is defined as $\mathcal{L}_{\text{est}} = \frac{1}{|B|}\sum_{b \in B}\|\mathbf{H}_{\text{pred}}^{(b)} - \mathbf{H}_{\text{gt}}^{(b)}\|_F^2$, where $B$ is a mini-batch of training samples. An L1 penalty on the base activations, $\mathcal{L}_{\text{activation}}$, encourages the model to use a sparse set of influential Gaussians and is given by $\mathcal{L}_{\text{activation}} = \frac{1}{N}\sum_{i=1}^N |\alpha_i|$. Finally, a regularization term, $\mathcal{L}_{\text{regularization}}$, prevents Gaussian primitives from becoming degenerate (either too large or too small) by constraining their scaling parameters $s_{i,j}$ within a predefined range $[s_{\min}, s_{\max}]$

$$\mathcal{L}_{\text{regularization}} = \frac{1}{N}\sum_{i=1}^N \sum_{j=1}^3 \left(\max(0, s_{\min} - s_{i,j}) + \max(0, s_{i,j} - s_{\max})\right). \tag{7}$$

### 3.5 GENERALIZATION

The explicit primitive field formulation extends beyond just radio fields by replacing the electromagnetic kernel $G_k$ with the Green's function for the target linear PDE while retaining the Gaussian sources and linear aggregator. In general form, $u(\mathbf{r}) = (G_{\mathcal{L}} * s)(\mathbf{r})$, $s(\mathbf{r}) \approx \sum_{i=1}^N \beta_i\, G_i(\mathbf{r}; \boldsymbol{\mu}_i, \boldsymbol{\Sigma}_i)$, where $G_{\mathcal{L}}$ is the Green's kernel of operator $\mathcal{L}$ (e.g., Helmholtz for acoustics, tensor Green's functions for elastodynamics, $1/\|\mathbf{r}\|$ for quasi-static Poisson problems). nGRF then learns per-primitive attributes appropriate to the field variable, while the spatial weight $w_i$ remains a Mahalanobis envelope. Because linear superposition and reciprocity are preserved in the aggregator, the model inherits the correct symmetries of the underlying physics, enabling the same accuracy-latency advantages and sample efficiency to carry over to other coherent sensing and propagation modalities.

## 4 EVALUATION

nGRF is evaluated across diverse propagation environments to demonstrate its generalizability. For indoor scenarios, three distinct environments are used: a conference room, a bedroom, and an office space. A large-scale residential area is used for the outdoor scenario. More details on the environments and the dataset generation process are provided in the Appendix B.

For antenna configurations, a $4 \times 4$ uniform rectangular array (URA) is used as the transmitter and a 2-element uniform linear array (ULA) as the receiver for indoor scenarios. For the outdoor scenario, the transmitter is scaled up to an $8 \times 8$ URA, while the receiver remains a 2-element ULA. To enable fair comparison with prior work, single-input single-output (SISO) setups with omnidirectional antennas are also configured. For each environment, 80% of the generated samples are used for training and 20% for testing. All experiments are implemented in PyTorch 2.7.0 with CUDA 12.8 bindings and trained on a single NVIDIA RTX 5090 GPU with 32 GB of memory.

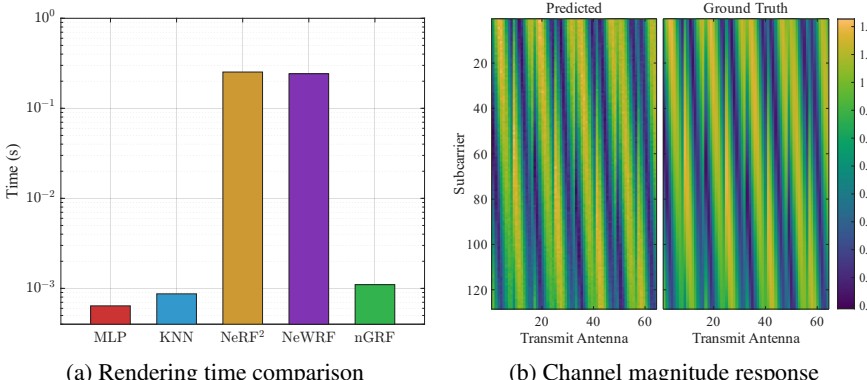

Figure 3: **Performance evaluation of nGRF.** (a) Rendering time comparison showing speedups of nGRF over other approaches. (b) Comparison between predicted (left) and ground-truth (right) channel magnitude response across subcarriers and transmit antennas in the outdoor environment.

**Results.** The performance of nGRF is compared against several baselines: NeWRF Lu et al. (2024), NeRF$^2$ Zhao et al. (2023), a standard multi-layer perceptron (MLP), and a k-nearest neighbors (KNN) approach. 3DGS-based methods are not included in this comparison, as they are designed to regress scalar power values (e.g., RSSI) or predict spatial spectra, rather than estimating the complex-valued CSI matrix which is the target of this work. Signal-to-noise ratio (SNR), defined as SNR (dB) $= 10 \log_{10}(\|\mathbf{H}_{gt}\|_F^2 / \|\mathbf{H}_{pred} - \mathbf{H}_{gt}\|_F^2)$, is used as the primary metric for evaluation.

**SNR Performance.** As shown in Table 1, nGRF consistently outperforms all baselines across all environments. In SISO configurations, it achieves an average SNR of 24.3 dB across indoor scenarios, representing a 10.9 dB improvement over the next-best method (NeWRF). In the large-scale outdoor scenario, where implicit methods struggle, nGRF achieves an SNR of 28.32 dB, while NeWRF and NeRF$^2$ fail to model the environment effectively. nGRF is also the only neural field method evaluated that supports MIMO configurations, maintaining high fidelity with an outdoor SNR of 27.92 dB. These results demonstrate that by embedding physical principles into the architecture, nGRF circumvents the typical accuracy-efficiency trade-off, achieving both superior performance and orders-of-magnitude speedups.

Table 1: **SNR (dB) across different scenarios.** The table compares nGRF against baselines for both SISO and MIMO configurations. Best results are in **bold**. NeRF-based methods do not support MIMO configurations.

| | Scenario | | | |
|---|---|---|---|---|
| **Method** | **Conference** | **Bedroom** | **Office** | **Outdoor** |
| *SISO Configuration* | | | | |
| MLP | -1.32 | -1.41 | -1.47 | 1.02 |
| KNN (k=5) | -2.25 | -2.37 | -2.32 | 0.95 |
| NeRF$^2$ | -0.44 | -1.22 | 0.77 | 1.40 |
| NeWRF | 21.64 | 12.38 | 4.96 | 2.03 |
| nGRF (ours) | **25.23** | **21.14** | **26.53** | **28.32** |
| *MIMO Configuration* | | | | |
| MLP | -1.98 | -1.99 | -2.11 | 1.81 |
| KNN (k=5) | -3.13 | -3.41 | -3.42 | 0.47 |
| NeRF$^2$ | – | – | – | – |
| NeWRF | – | – | – | – |
| nGRF (ours) | **22.73** | **18.60** | **24.78** | **27.92** |

Table 2: **Comparison of data and computational efficiency.** The table summarizes training metrics for the indoor (conference) scenario.

| Method | SNR (dB) | Train Time | Measurement Density (samples/ft$^3$) |
|---|---|---|---|
| MLP | -1.32 | <1 min | 178.1 |
| KNN (k=5) | -2.25 | – | 178.1 |
| NeRF$^2$ | -0.44 | ∼5 hours | 178.1 |
| NeWRF | 21.64 | ∼2.43 hours | 0.2 |
| **nGRF (ours)** | **25.23** | **2.3 min** | **0.011** |

In addition to accuracy, the practical viability of any method depends on its computational and data efficiency. Table 2 provides an overview of these resources for the indoor conference room scenario. The measurement densities for the NeRF-based methods reflect the high data requirements needed to achieve their maximum reported SNR. To create the strongest possible baseline for data-driven methods, MLP and KNN are provided with the highest available measurement density (178.1 samples/ft$^3$), which matches that of NeRF$^2$. Despite this, their performance remains poor and showcases the limitations of physics-agnostic models. In contrast, nGRF achieves superior accuracy while requiring $18\times$ less data than NeWRF and over $16,000\times$ less than NeRF$^2$.

**Rendering Time.** Figure 3a compares the inference latency of the different methods. nGRF achieves channel estimation in just 1.1 ms, a $220\times$ speedup compared to 242 ms for NeWRF and 253 ms for NeRF$^2$. While MLP and KNN are marginally faster, their poor accuracy makes them impractical. For dynamic environments with channel coherence times as short as 2 ms, the combination of high accuracy and low latency makes nGRF the only viable solution among the tested methods. The performance gap highlights a key architectural difference: implicit models require hundreds of expensive MLP queries per estimate, whereas nGRF's explicit, physics-informed basis enables direct computation.

**Frequency Generalization.** A key capability of nGRF is its ability to generalize across frequencies. Figure 3b shows the channel magnitude response across all subcarriers for a specific receiving antenna in the outdoor environment. The model was trained using data from only a single subcarrier (subcarrier 64), yet it accurately predicts the channel response across the entire frequency band. This is possible because nGRF learns a representation of the underlying spatial structure of the electromagnetic field, which is governed by the environment's geometry and is largely frequency-agnostic within the coherence bandwidth. Physical path delays translate to predictable linear phase shifts across frequencies, which is a relationship that the model implicitly captures. Thus, nGRF is not merely fitting to frequency-specific patterns but rather is learning a physical model of the environment in the context of electromagnetic wave propagation.

**Ablation Studies.** Several ablation studies are performed to understand the contribution of different components of nGRF, with results for the outdoor environment summarized in Table 3.

1. *Number of Gaussians.* Reducing the number of Gaussians to 500 or 1,000 maintains competitive performance. However, increasing this figure to 5,000 or 10,000 leads to severe degradation. The experiments in Table 3 indicate that while a minimum threshold of Gaussians is necessary to capture electromagnetic field complexity, excessive parameterization induces overfitting and diminishing returns.

2. *Gaussian Positions.* When the positions of Gaussian primitives are fixed (made non-trainable), the SNR drops from 28.32 dB to 4.11 dB. Optimizing the spatial distribution of Gaussians is crucial for accurately modeling the electromagnetic field, allowing the model to adapt to the specific propagation characteristics of the environment.

3. *Initialization Strategy.* Contrary to intuition, initializing Gaussian means using a LiDAR-generated point cloud yields worse performance than random initialization. In nGRF, Gaussians function not as physical scatterers of radio waves but as adaptive basis functions for the radio propagation field. Consequently, geometry-constrained initialization restricts the model's ability to capture complex wave phenomena that transcend environmental geometry.

Additional ablation studies are provided in Appendix C.

Table 3: **Ablation experiments for nGRF.** Performance variations across different numbers of Gaussians, optimization strategies, and initialization methods are evaluated for the outdoor scenario.

| Configuration | SNR (dB) | Train Time (min) | Render Time (ms) |
|---|---|---|---|
| nGRF (baseline) | 28.32 | 2.3 | 1.10 |
| *# of Gaussians* | | | |
| 500 | 26.13 | 1.9 | 0.94 |
| 1,000 | 26.57 | 2.2 | 1.06 |
| 5,000 | 23.31 | 3.0 | 1.27 |
| 10,000 | 18.04 | 3.6 | 1.40 |
| *Positions $\boldsymbol{\mu}_i$* | | | |
| Fixed means | 4.11 | 2.3 | 1.10 |
| *Initialization* | | | |
| LiDAR-based | 19.76 | 2.3 | 1.10 |

## 5 Limitations and Future Work

Despite its advantages, nGRF exhibits two notable **limitations**. First, the framework demonstrates significant hyperparameter sensitivity, particularly to Gaussian scaling parameters and initialization values; performance can degrade by up to 14.49 dB with suboptimal scaling initialization, with similar sensitivity observed for learning rates and regularization coefficients. Second, the current formulation lacks mechanisms for handling time-varying channels in dynamic environments with mobile users.

**Future work** could focus on developing robust hyperparameter selection through meta-learning approaches and extending nGRF to model temporal dynamics through time-dependent Gaussian attributes. Furthermore, the implementation of incremental feedback mechanisms, where channel updates are based on prediction errors, could enable real-time adaptation in mobility scenarios. This would facilitate predictive channel estimation for high-velocity applications where coherence times approach the millisecond threshold.

## 6 Conclusion and Broader Impact

This work introduced nGRF, a framework that synthesizes complex MIMO channel matrices by directly aggregating explicit 3D Gaussian primitives, each acting as a learned radio modulator. nGRF achieves state-of-the-art channel estimation accuracy with major reductions in latency, training time, and data requirements, overcoming key limitations of prior implicit and projection-based methods. The core contribution lies in demonstrating that structured, explicit representations, informed by physical principles, can provide a better way of modeling complex field phenomena than plain function-learning deep learning architectures.

The underlying principle, i.e., representing a field as a sum of explicit, localized sources governed by a physics-based aggregation rule, is highly generalizable. By substituting the model for electromagnetic propagation with the appropriate kernels or Green's functions, this approach could be adapted to create primitive-based neural fields for acoustics, elastodynamics, or diffusion phenomena. Beyond wireless communications, the principles demonstrated in nGRF offer a blueprint for developing efficient neural field models in other scientific and engineering domains where capturing complex interactions is necessary. This methodology has the potential to accelerate discovery while reducing the carbon footprint of large-scale AI by enabling more sample- and computationally efficient modeling.

## ETHICS STATEMENT

The authors have read and adhered to the ICLR Code of Ethics. This work focuses on a fundamental problem in wireless communications and does not involve human subjects, sensitive personal data, or applications with immediate potential for harm. The datasets used for evaluation were synthetically generated using standard ray-tracing techniques, as detailed in the appendix, and do not contain any personally identifiable information. To improve the clarity, writing, and formatting of this paper, commercial large language models (LLMs) were utilized as an assistive tool, specifically ChatGPT by OpenAI and Claude by Anthropic.

## REPRODUCIBILITY STATEMENT

All code and datasets used in the experiments are made publicly available in an anonymized repository at `https://github.com/anonym-auth/n-grf` to ensure reproducibility of the presented results. The repository includes instructions for installing and compiling all necessary dependencies, including the custom CUDA kernels. Furthermore, scripts for training the nGRF model and evaluating it against the baselines are also provided. The appendix of this paper contains a thorough description of the dataset generation process, antenna configurations, and specific hyperparameters used for the experiments, which, in conjunction with the provided code, should enable the reproduction of our findings.

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

APPENDIX

# A  MIMO PRELIMINARIES

This section recalls a standard narrowband multipath and array formulation for wireless channels and connects it to pilot-based estimators used in practice. Throughout, notation maintains consistency by identifying $N_{\text{tx}} \equiv N_t$ and $N_{\text{rx}} \equiv N_r$ across the modeling and estimation subsections.

## A.1  PROPAGATION AND ARRAY MODEL

A complex baseband symbol is $x = Ae^{j\psi}$. Over distance $d$ at carrier frequency $f$, free-space loss and phase evolve as Rappaport et al. (2013)

$$A_{\text{att}}(d) = \frac{c}{4\pi d\,f}, \qquad \Delta\psi(d) = -\frac{2\pi f\,d}{c}. \tag{8}$$

Reflections, diffractions, and penetrations induce path-dependent gains and phases Rappaport et al. (2013); Goldsmith (2005). With $L$ paths, the received symbol is

$$y = \sum_{l=0}^{L-1} A_l\, A_{\text{att},l}\, e^{j(\psi + \Delta\psi_l)}, \tag{9}$$

and the (SISO) channel is the complex ratio

$$h = \frac{y}{x} = \sum_{l=0}^{L-1} A_{\text{att},l}\, e^{j\Delta\psi_l}. \tag{10}$$

For an $N_{\text{rx}} \times N_{\text{tx}}$ array, steering vectors aggregate per-path effects:

$$\mathbf{H} = \sum_{l=0}^{L-1} \alpha_l\, \mathbf{a}_r(\vartheta_l^r)\, \mathbf{a}_t^H(\vartheta_l^t), \qquad \alpha_l = A_{\text{att},l}\, e^{j\Delta\psi_l}. \tag{11}$$

Here $\mathbf{a}_t(\vartheta_l^t)$ and $\mathbf{a}_r(\vartheta_l^r)$ are transmit and receive steering vectors at departure and arrival angles $\vartheta_l^t, \vartheta_l^r$, respectively.

3GPP TR 38.901 clustered delay-line (CDL) models randomize $(\alpha_l, \tau_l, \vartheta_l^t, \vartheta_l^r)$ across standardized scenarios from $0.5$–$100\,\text{GHz}$ for 5G/6G evaluation. At mmWave frequencies ($f > 24\,\text{GHz}$), atmospheric absorption and rain add frequency-selective losses that shorten viable link distances Raleigh & Cioffi (1998).

## A.2  PILOT-BASED MIMO CHANNEL ESTIMATION

For a narrowband $N_t \times N_r$ MIMO link, pilots are inserted so that the receiver observes

$$\mathbf{Y} = \mathbf{H}\mathbf{X} + \mathbf{N},$$

where $\mathbf{H} \in \mathbb{C}^{N_r \times N_t}$ is the (per-subcarrier) channel, $\mathbf{X} \in \mathbb{C}^{N_t \times T}$ is the known pilot matrix sent over $T$ symbol periods, and $\mathbf{N} \sim \mathcal{CN}(\mathbf{0}, \sigma_n^2\mathbf{I})$ is additive white Gaussian noise Salz & Winters (2002). In practice, $\mathbf{X}$ is designed to be *orthogonal*, i.e.,

$$\mathbf{X}\mathbf{X}^H = \rho\,\mathbf{I}_{N_t},$$

which decouples the transmit streams and makes inversion stable when $T \geq N_t$ Yun et al. (2025). The receiver then estimates $\mathbf{H}$ via least squares:

$$\hat{\mathbf{H}}_{\text{LS}} = \arg\min_{\mathbf{H}} \|\mathbf{Y} - \mathbf{H}\mathbf{X}\|_F^2 \;=\; \mathbf{Y}\mathbf{X}^H(\mathbf{X}\mathbf{X}^H)^{-1} \;=\; \frac{1}{\rho}\,\mathbf{Y}\mathbf{X}^H.$$

This LS estimator is simple, per-subcarrier parallelizable, and widely used; its accuracy improves with pilot SNR and orthogonality, while increased pilot density trades spectral efficiency for lower estimation error.

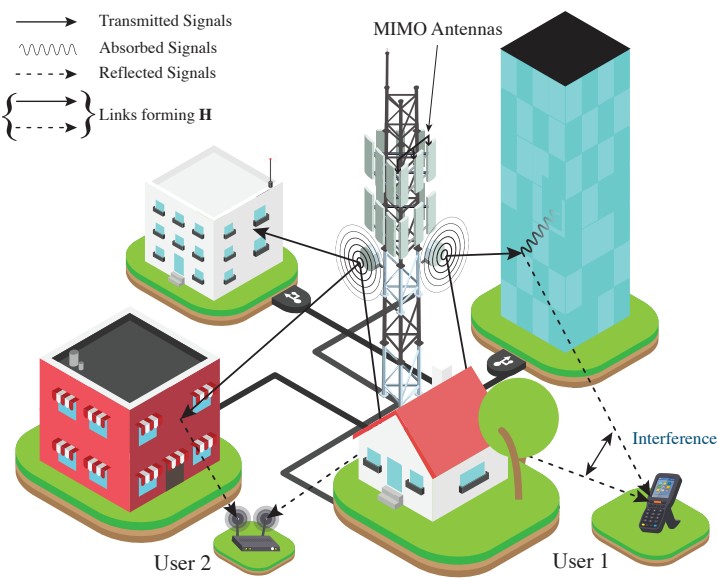

Figure 4: Multipath propagation with path-dependent attenuation and phase.

## B    DATASET GENERATION

Physics-consistent channels are constructed via geometry-based ray tracing. STL models (indoor: conference room, bedroom; outdoor: urban block) provide vertices, faces, and material tags. A URA transmitter and randomly placed receivers define communication links; propagation is simulated using MATLAB `RayTracing`, yielding path sets for each link.

For each ray of length $d$ at carrier frequency $f$, free-space path loss (in dB) and phase are computed as

$$\text{FSPL} = 20 \log_{10} d + 20 \log_{10} f + 20 \log_{10}\left(\frac{4\pi}{c}\right), \qquad \phi = -2\pi f \tau, \ \ \tau = \frac{d}{c}. \tag{12}$$

Reflections and diffractions add material- and angle-dependent losses. The Fresnel reflection coefficients for incidence angle $\theta_i$ and complex permittivity $\varepsilon_r$ are given by

$$\Gamma_p = \frac{\sin\theta_i - \sqrt{\varepsilon_r - \cos^2\theta_i}}{\sin\theta_i + \sqrt{\varepsilon_r - \cos^2\theta_i}}, \qquad \Gamma_s = \frac{\varepsilon_r \sin\theta_i - \sqrt{\varepsilon_r - \cos^2\theta_i}}{\varepsilon_r \sin\theta_i + \sqrt{\varepsilon_r - \cos^2\theta_i}}. \tag{13}$$

Array geometry is embedded via steering vectors. For a URA transmitter with element positions $\mathbf{d}_T$ and azimuth/elevation angles $(\alpha_T, \beta_T)$,

$$\mathbf{a}_T\big(f, [\alpha_T; \beta_T]\big) = \exp\left(j\frac{2\pi f}{c}\,\mathbf{d}_T \cdot [\cos\alpha_T \cos\beta_T, \ \sin\alpha_T \cos\beta_T, \ \sin\beta_T]^T\right), \tag{14}$$

and similarly for $\mathbf{a}_R$ at the receiver array. For the $l$-th path with amplitude $a_l$ (including FSPL, Fresnel, and UTD factors) and phase $\phi_l$,

$$\mathbf{H}_l = a_l e^{j\phi_l}\,\mathbf{a}_R\,\mathbf{a}_T^H, \qquad \mathbf{H} = \sum_{l=0}^{L-1} \mathbf{H}_l \ \in \mathbb{C}^{N_r \times N_t}. \tag{15}$$

For SISO configurations, the channel reduces to the scalar superposition

$$h = \sum_{l=0}^{L-1} a_l e^{j\phi_l}. \tag{16}$$

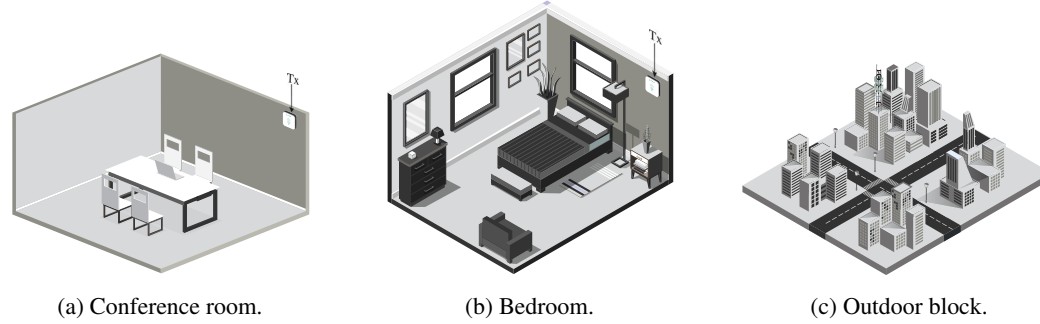

(a) Conference room.      (b) Bedroom.      (c) Outdoor block.

Figure 5: 3D environments used for geometry-based ray tracing.

Table 4: **Hyperparameter sensitivity analysis.** The impact of various hyperparameters on model performance is evaluated, including learning rates, batch size, activation regularization, and position optimization cutoff.

| Configuration | SNR (dB) | Train Time (min) | Convergence Iter. |
|---|---|---|---|
| nGRF (baseline) | 25.23 | 2.3 | 2841 |
| *Position Learning Rate* | | | |
| $\eta_{\text{pos}} = 0.0005$ | 18.47 | 2.4 | 3152 |
| $\eta_{\text{pos}} = 0.001$ | 20.94 | 2.3 | 2974 |
| $\eta_{\text{pos}} = 0.005$ | 23.16 | 2.2 | 2753 |
| $\eta_{\text{pos}} = 0.01$ | 22.84 | 2.1 | 2611 |
| *Batch Size* | | | |
| 8 | 21.78 | 2.0 | 3102 |
| 16 | 22.45 | 2.1 | 2937 |
| 64 | 23.18 | 2.5 | 2783 |
| 128 | 22.96 | 3.1 | 2901 |
| *L1 Activation Regularization* | | | |
| $\lambda_{\text{act}} = 0.0$ | 21.37 | 2.3 | 2945 |
| $\lambda_{\text{act}} = 0.05$ | 22.84 | 2.3 | 2887 |
| $\lambda_{\text{act}} = 0.2$ | 22.17 | 2.2 | 2904 |
| *Position Update Cutoff* | | | |
| No cutoff | 22.14 | 2.4 | 3076 |
| 30% iterations | 20.86 | 2.3 | 3184 |
| 80% iterations | 22.95 | 2.3 | 2798 |

## C ABLATION STUDIES

This section presents additional ablation studies that assess the sensitivity of nGRF to various hyperparameters and design choices. Note that these experiments are intended to complement the ablations presented in the main paper and provide further insights into the model's behavior.

### C.1 HYPERPARAMETER SENSITIVITY

The sensitivity of nGRF to key hyperparameters, including learning rates, batch size, and regularization factors, is assessed. Table 4 summarizes the findings across different hyperparameter configurations in the conference room environment.

Table 5: **Impact of Gaussian scaling parameters.** The analysis covers how different initial scaling values and constraints affect model performance across multiple environments.

| Configuration | Indoor SNR (dB) | Outdoor SNR (dB) | Render Time (ms) |
|---|---|---|---|
| nGRF (baseline, $s_{\text{init}} = 0.137$) | 25.23 | 28.32 | 1.10 |
| *Initial Scale Value* | | | |
| $s_{\text{init}} = 0.001$ | 14.47 | 17.76 | 1.08 |
| $s_{\text{init}} = 0.005$ | 13.83 | 18.42 | 1.09 |
| $s_{\text{init}} = 0.01$ | 14.21 | 19.05 | 1.09 |
| $s_{\text{init}} = 0.05$ | 24.74 | 19.87 | 1.12 |
| $s_{\text{init}} = 0.1$ | 25.23 | 28.32 | 1.15 |
| $s_{\text{init}} = 0.2$ | 22.91 | 27.85 | 1.18 |
| *Scale Constraints* | | | |
| Unconstrained | 20.32 | 23.18 | 1.21 |
| Tight ($s \in [0.05, 0.2]$) | 24.04 | 27.95 | 1.07 |
| Wide ($s \in [0.001, 0.5]$) | 22.76 | 24.91 | 1.13 |

Optimal performance is achieved with a position learning rate of 0.005, matching the baseline configuration. Lower learning rates appear to lead to insufficient spatial exploration, while higher rates can destabilize training. Batch sizes between 32 and 64 provide a good trade-off between convergence speed[1] and generalization. Furthermore, L1 regularization on activations ($\lambda_{\text{act}}$) promotes sparsity; without it ($\lambda_{\text{act}} = 0.0$), performance degrades by approximately 1.9 dB.

Regarding the position update cutoff, the point at which updates to Gaussian positions are halted, the results indicate that continuing position updates for 60-65% of training iterations (as in the baseline) allows the Gaussians to settle into optimal locations before their attributes are fine-tuned. Disabling this cutoff entirely leads to an SNR drop of 1.1 dB, likely because the model struggles to optimize positions and attributes simultaneously.

## C.2   GAUSSIAN SCALING

The effect of initialization and constraints on Gaussian scaling parameters is examined to understand their impact on the model's ability to represent the electromagnetic field. Table 5 presents the experimental results with different scaling configurations.

The results indicate that nGRF is highly sensitive to Gaussian scaling parameters. For indoor environments, initial scaling values between 0.05 and 0.2 consistently yield good performance, while very small values perform poorly. The outdoor environment shows a preference for larger scaling values, with optimal performance observed when $s_{\text{init}}$ is between 0.1 and 0.2. This behavior is attributed to the nature of nGRF's spatial weighting. Since the weight depends on the Gaussian's covariance, smaller Gaussians have a more localized influence. Achieving a good fit with small Gaussians would require a significantly larger number of them, which can lead to overfitting. Consequently, using larger Gaussians that can cover more area and capture the overall field distribution is more effective.

Constraining scale parameters during training also improves performance, particularly when the constraints align with the optimal scale ranges for each environment. In these experiments, tight constraints centered around the optimal ranges ($s \in [0.05, 0.2]$) maintain high SNR while reducing variance. Unconstrained scales lead to performance degradation of approximately 3.1 dB indoors and 3.25 dB outdoors, as the Gaussians may converge to suboptimal scales.

---

[1]Convergence is defined as the point at which the model achieves the highest SNR on the validation set and does not degrade by more than 1 dB over the next 500 iterations.

Table 6: **Impact of positional encoding frequency bands.** The evaluation shows how the number of frequency bands affects the model's ability to capture spatial variations in the electromagnetic field.

| Configuration | SNR (dB) | Train Time (min) |
|---|---|---|
| nGRF (baseline, $L = 16$) | 25.23 | 2.3 |
| *Frequency Bands* | | |
| $L = 4$ | 21.07 | 2.0 |
| $L = 8$ | 23.83 | 2.1 |
| $L = 10$ | 23.95 | 2.1 |
| $L = 12$ | 24.05 | 2.2 |
| $L = 16$ | 25.23 | 2.3 |
| $L = 32$ | 23.35 | 2.4 |

Table 7: **Impact of dataset characteristics.** The evaluation covers how measurement density and noise levels affect model performance.

| Configuration | SNR (dB) | Train Time (min) |
|---|---|---|
| nGRF (baseline) | 25.23 | 2.3 |
| *Measurement Density (samples/ft$^3$)* | | |
| 0.005 | 19.47 | 2.0 |
| 0.01 | 24.56 | 2.2 |
| 0.02 | 25.01 | 2.4 |
| 0.05 | 24.51 | 2.7 |
| *Noise Level* | | |
| No noise ($\sigma = 0$) | 21.34 | 2.2 |
| Low noise ($\sigma = 0.001$) | 24.85 | 2.2 |
| Baseline ($\sigma = 0.00387$) | 25.23 | 2.3 |
| Med noise ($\sigma = 0.02$) | 23.67 | 2.4 |
| High noise ($\sigma = 0.1$) | 18.42 | 2.8 |

## C.3 POSITIONAL ENCODING

The frequency of the positional encoding affects the model's capacity to capture high-frequency variations in the electromagnetic field. Table 6 shows the effect of using different numbers of frequency bands in the positional encoding on model performance.

The experiments show that nGRF's performance is relatively insensitive to the specific number of positional encoding frequency bands, provided it is above a minimum threshold. When the number of frequency bands is very low ($L = 4$), a performance decrease of 4.16 dB compared to the baseline is observed, indicating insufficient capacity to represent high-frequency spatial variations.

Beyond this threshold, however, performance remains remarkably stable. The SNR varies by at most 1-2 dB across all configurations from $L = 8$ to $L = 32$, with no clear monotonic improvement as the number of frequency bands increases. Performance peaks at $L = 16$ and declines slightly with $L = 32$, which suggests that excessive frequency bands may introduce unnecessary high-frequency components that can lead to overfitting.

## C.4 MEASUREMENT DENSITY & NOISE

This subsection analyzes how various dataset characteristics affect nGRF's performance, focusing on measurement density and robustness to noise. Table 7 summarizes these findings.

With just 0.01 samples/ft$^3$, nGRF achieves an SNR of 24.56 dB, only 0.67 dB below the baseline performance. Doubling the density to 0.02 samples/ft$^3$ yields only a minor improvement, while further increases result in a slight, negligible degradation in SNR. This result suggests that nGRF's structured Gaussian representation effectively interpolates between sparse measurements, allowing it to achieve high fidelity with $18\times$ fewer measurements than comparable neural field approaches.

Consistent with prior work in neural fields, a certain level of noise during training is found to be beneficial. Training without noise ($\sigma = 0$) results in an SNR 3.89 dB below the baseline. A small amount of positional noise (with $\sigma = 0.00387$ in the baseline) acts as a regularizer, preventing overfitting and encouraging the model to learn smoother, more generalizable field representations.

Excessive noise, however, is problematic. Medium noise levels ($\sigma = 0.02$) reduce performance by 1.56 dB, while high noise ($\sigma = 0.1$) causes a significant degradation of 6.81 dB. This highlights the existence of an optimal noise level and underscores the model's sensitivity to this hyperparameter, which is a notable limitation of nGRF.