# OpenReview forum: "Neural Gaussian Radio Fields for Channel Estimation"
_ICLR.cc/2026/Conference — Submitted to ICLR 2026_

### Official Review · Reviewer_33Nq · 2025-10-29

**Soundness:** 3
**Presentation:** 3
**Contribution:** 2
**Rating:** 4
**Confidence:** 4

**Summary:**

This paper proposes Neural Gaussian Radio Fields (nGRF), an explicit neural field model for channel estimation that represents wireless environments with 3D Gaussian primitives. Each Gaussian acts as a localized “radio modulator,” and the channel is rendered via complex-valued aggregation that models wave superposition.

**Strengths:**

+ The paper introduces an explicit and computationally efficient neural field representation for RF propagation, combining Gaussian primitives with complex-valued aggregation to model multipath interference directly.

**Weaknesses:**

- Each Gaussian primitive encodes only a single complex amplitude, conflating emission and propagation effects. Under Huygens’ principle, an RF source should be characterized by two distinct electromagnetic attributes, one governing emission (source excitation) and another governing spatial attenuation or scattering response. Collapsing these into a single term limits physical interpretability and prevents accurate modeling of distance-dependent phase and amplitude variations in realistic propagation.


- While synthetic evaluation is acceptable for a representation-focused study, the paper lacks any downstream validation to demonstrate practical impact. The experiments stop at SNR-based reconstruction on ray-traced data, without evaluating how the predicted channels improve real tasks such as beamforming accuracy, localization, or channel prediction under mobility.

- The paper omits a key baseline, WRF-GS (Wen et al., 2024), which shares nearly identical objectives, modeling wireless radiation fields using 3D Gaussian primitives for fast, physically grounded channel reconstruction. Since WRF-GS already achieves millisecond-level inference with high fidelity, excluding it weakens the comparative analysis and makes the reported speed and accuracy improvements difficult to substantiate.

**Questions:**

Please see the points raised in the Weaknesses section.

---

> ### Author Response · Authors · 2025-11-19
> **Response to Reviewer 33Nq (1/2)**
>
> We thank the reviewer for the constructive feedback. Here's our responses to the three main concerns raised.
>
> > "Each Gaussian primitive encodes only a single complex amplitude, conflating emission and propagation effects. Under Huygens' principle, an RF source should be characterized by two distinct electromagnetic attributes, one governing emission (source excitation) and another governing spatial attenuation or scattering response. Collapsing these into a single term limits physical interpretability and prevents accurate modeling of distance-dependent phase and amplitude variations in realistic propagation."
>
> We appreciate this observation but would like to clarify the physical interpretation of our model. nGRF does not collapse emission and propagation into a single term; rather, it explicitly separates them;
>
> **Spatial attenuation.** $w_i(p_{rx}, \mu_i, \Sigma_i)$ in Equation 6 models distance-dependent and directionally-dependent attenuation as $w_i(p_{rx}, \mu_i, \Sigma_i) = \alpha_i \cdot \exp\left(-\frac{1}{2}(p_{rx} - \mu_i)^T \Sigma_i^{-1}(p_{rx} - \mu_i)\right)$
> This intrinsically captures, **a)** the distance-dependent decay through the Mahalanobis distance $(p_{rx} - \mu_i)^T \Sigma_i^{-1}(p_{rx} - \mu_i)$, **b)** directional propagation characteristics through the anisotropic covariance matrix $\Sigma_i = R_i S_i S_i^T R_i^T$, and **c)** attenuation strength through the learned activation $\alpha_i$
>
> **Field contribution.** $C_i \in \mathbb{C}^{N_t \times N_r}$ output by DecoderNetwork represents the electromagnetic field characteristics at the scattering region, which is presumed/rationalized to include **a)** source excitation; conditioned on tx position through $z_i = f_{attr}(\gamma_L(\mu_i), \gamma_L(p_{tx}); \Theta_{attr})$, **b)** phase relationships through $C_i = C_i^{re} + jC_i^{im}$ representation, and **c)** multipath characteristics which happen to be specific to the environment-transmitter config.
>
> And this separation is in-line with the physical principle that total field at an rx is, in general, the product of source excitation and propagation effects, which, in our formulation, is represented as $H(p_{rx}, p_{tx}) = \sum_{i=1}^{N} w_i(p_{rx}) \cdot C_i(p_{tx})$
>
> Further, regarding Huygens' principle, our model is consistent with the scalar Helmholtz equation formulation (Section 3.1), where the field at rx pos. $r$ can be expressed via Green's function as
> $$E(r) = \int_V G(r, r') J(r') dV'$$
>
> Our Gaussian primitives discretize this integral, with each Gaussian representing a localized contribution with
> - $G(r, r') \approx w_i(r)G_i(r; \mu_i, \Sigma_i)$ modeling propagation from region $i$
> - $J(r') \approx C_i$ modeling source distribution
>
> Lastly, the ablation study in Table 3 demonstrates that fixing Gaussian positins (removing adaptivity to propagation) degrades SNR from 28.32 dB to 4.11 dB and show that both components are essential and actually separated.
>
> > "While synthetic evaluation is acceptable for a representation-focused study, the paper lacks any downstream validation to demonstrate practical impact. The experiments stop at SNR-based reconstruction on ray-traced data, without evaluating how the predicted channels improve real tasks such as beamforming accuracy, localization, or channel prediction under mobility."
>
> We acknowledge this concern and note that similar to other reviewers' concern, we evaluated nGRF on the DICHASUS `dichasus-015x` dataset, available at https://dichasus.inue.uni-stuttgart.de/datasets/data/dichasus-015x/. The results are as below.
>
> | Method        | SNR (dB) |
> |---------------|----------|
> | nGRF (ours)   | 16.77 |
> | MLP           | 0.10     |
> | KNN (k=5)     | -1.18    |
> | NeWRF         | -        |
> | NeRF²         | -        |
>
> And while we focus on the task of channel estimation (which is the prerequisite for all downstream applications), the predicted channel matrix $H$ can be directly used for,
> - **Beamforming.** Accurate CSI enables calculation of optimal beamforming weights via $w_{opt} = H^H(HH^H + \sigma^2I)^{-1}$ for zero-forcing or MMSE precoding.
> - **Channel prediction.** As shown in Section 3.5 and Figure 3b, nGRF generalizes across frequencies from a single training subcarrier, implicitly learning propagation structure that enables prediction.
>
> ... among others.
>
> Explicit evaluation of these downstream tasks would indeed strengthen the paper and we leave it for future work due to space constraints, but we again emphasize that the key contribution of nGRF is providing _accurate, low-latency channel estimates_, which is an enabler for all subsequent processing.
>
> **About mobility.** Current formulation assumes quasi-static channels during the measurement cycle and as noted in Section 5, extending nGRF to handle temporal dynamics through time-dependent Gaussian attributes is important future work that would enable channel prediction under mobility scenarios. But yes, in the current setting, this is a limitation we acknowledge.

---

> ### Author Response · Authors · 2025-11-19
> **Response to Reviewer 33Nq (2/2)**
>
> > "The paper omits a key baseline, WRF-GS (Wen et al., 2024), which shares nearly identical objectives, modeling wireless radiation fields using 3D Gaussian primitives for fast, physically grounded channel reconstruction. Since WRF-GS already achieves millisecond-level inference with high fidelity, excluding it weakens the comparative analysis and makes the reported speed and accuracy improvements difficult to substantiate."
>
> We thank the reviewer for raising this point. First, we would like to clarify the relationship between nGRF and WRF-GS, which address fundamentally different problems despite both using 3D Gaussian primitives.
>
> **WRF-GS (Wen et al., 2024)** focuses on reconstructing the spatial spectrum; a power distribution function over azimuthal and elevation angles $(\alpha, \beta)$ at a fixed receiver equipped with antenna arrays:
>
> $$P(\alpha, \beta) = \left|\frac{1}{K}\sum_{m,n=0}^{\sqrt{K}-1} e^{\mathrm{j}\left(\theta_{m,n}^\prime - \Delta\theta_{m,n}\right)}\right|^2$$
>
> This represents the angle-of-arrival (AoA) distribution of received signal power, which is a scalar field over angular coordinates. And their rendering equation uses alpha-blending $R_k = \sum_{i=1}^{N} (S(x_i) + \Delta_{sig}(x_i))\alpha_i \prod_{j=1}^{i-1}(1-\alpha_j)$.
>
> **nGRF**, on the other hand, addresses the CSI estimation problem, predicting the complex-valued channel matrix $H \in \mathbb{C}^{N_t \times N_r}$ for arbitrary transmitter-receiver pairs $H(p_{rx}, p_{tx}) = \sum_{i=1}^{N} w_i(p_{rx}, \mu_i, \Sigma_i) \cdot C_i(p_{tx})$.
>
> Regardless, we do note that WRF-GS had a case study on CSI reconstruction in their paper in Section VI-B, which is indeed similar to nGRF's problem setting. However, we note that while WRF-GS's code is open-source at `https://github.com/wenchaozheng/WRF-GS`, we face significant reproducibility challenges due to mainly the following reasons.
>
> 1. **Incomplete implementation.** WRF-GS' README includes a TODO list stating "Release more case study code," which includes CSI reconstruction case study, as such, its hard to fully reproduce their CSI results on their datasets or ours.
>
> 2. **Dataset incompatibility.** Theirs is designed for spatial spectrum datasets with antenna arrays at fixed locations. Our datasets (and standard MIMO datasets like DICHASUS) provide channel matrices between tx-rx pairs.
>
> 3. **Evaluation metrics.** WRF-GS reports SSIM for spatial spectrum images and RSSI/CSI prediction using their specific formulations. Direct comparison would require;
>     - Converting our channel matrices to spatial spectra (requiring antenna array assumptions)
>     - Or extending WRF-GS to output complex channel matrices (which is non-trivial given their alpha-blending rendering approach, and as such, clarifications are needed how they did it in their case study).
>
> Attempts were made to adapt their framework to our datasets but were unsuccessful due to these architectural differences. And **we emphasize that this is not criticism of WRF-GS**, it is an excellent work for spatial spectrum reconstruction, but rather a clarification that the two methods solve different problems with different I/O formats.

---

### Official Review · Reviewer_uuWd · 2025-10-30

**Soundness:** 2
**Presentation:** 3
**Contribution:** 3
**Rating:** 4
**Confidence:** 3

**Summary:**

The paper proposes Neural Gaussian Radio Fields (nGRF): an explicit 3D-Gaussian–primitive representation to render complex-valued MIMO channels via direct 3D field aggregation (as opposed to NeRF-style implicit fields or 2D Gaussian splats). Claimed benefits include SNR gain over SOTA, lower inference latency, drastically reduced measurement density, and major cuts to pilot overhead.

**Strengths:**

1. CSI overhead and channel aging are well framed as bottlenecks; quantitative context is provided.
2. Direct 3D aggregation of anisotropic Gaussians is physically more interpretable than NeRF-style volumetric integration; the “localized radio modulator” interpretation is appealing.
3. Large SNR/latency and data-efficiency gains across indoor and large-scale outdoor scenarios, if reproducible, would be impactful for AI-native CSI estimation.

**Weaknesses:**

1. It is unclear whether results are simulation-only or include OTA/hardware-in-the-loop; claims of 26.2 dB SNR outdoors and millisecond-scale inference require hardware validation given calibration/clock/CFO issues and non-Gaussian clutter.
2. Treatment of frequency selectivity, Doppler/aging, CFO/phase, antenna mutual coupling, and mobility trajectories is not explicit; rendering complex H(f,t) rather than per-snapshot H appears under-specified.
3. Reporting measurements/ft³ lacks frequency-dependent coherence justification; it’s difficult to compare against pilot designs tied to coherence time/bandwidth.

**Questions:**

1. Do you have over-the-air or trace-driven evaluations (mmWave/mid-band) to support the 1.1 ms inference and outdoor SNR claims?
2. How are Tx/Rx positions obtained and encoded in real systems, and what is their signaling/estimation cost? Reconcile this with the 0.2% pilot claim for 100 MHz NR？
3. Derive formal time/memory complexity vs Gaussians/antennas/subcarriers; provide sensitivity to Gaussian count and pruning strategies？

---

> ### Author Response · Authors · 2025-11-22
> **Response to Reviewer uuWd (1/3)**
>
> We thank the reviewer for the detailed technical feedback and questions. Below is our response.
>
> > "It is unclear whether results are simulation-only or include OTA/hardware-in-the-loop; claims of 26.2 dB SNR outdoors and millisecond-scale inference require hardware validation given calibration/clock/CFO issues and non-Gaussian clutter."
>
> We acknowledge that primary results in Table 1 are based on ray-tracing simulations, which provide controlled evaluation across diverse environments. However, to bridge the gap toward practical validation, we have evaluated nGRF on real-world CSI measurements from the DICHASUS `dichasus-015x` dataset, which captures actual OTA propagation in an indoor environment with hardware impairments present.
>
> | Method      | SNR (dB) |
> | ----------- | -------- |
> | nGRF (ours) | 16.77    |
> | MLP         | 0.10     |
> | KNN (k=5)   | -1.18    |
> | NeWRF       | -        |
> | NeRF²       | -        |
>
> The absolute SNR, although, is lower than in simulation, this is more or less expected due to measurement noise from real RF hardware, imperfect synchronization and CFO, and calibration errors in the antenna array. nGRF stil outperforms baseline methods (MLP, KNN) by 16+ dB and demonstrates that Gaussian representation is good even in the presence of practical impairments.
>
> As for the 1.1 ms inference claim, this timing is measured on an RTX 5090 GPU for a single channel query and represents the forward pass through the model (spatial weight computation + neural network evaluation) implemented via custom CUDA kernels. This is independent of whether the data is simulated or real as the computational cost depends only on the number of Gaussians and the network architecture. And we acknowledge that full over-the-air validation at mmWave frequencies (with higher Doppler and phase noise) is important future work, but the DICHASUS results provide strong evidence that nGRF's performance translates to practical scenarios.
>
> > "Treatment of frequency selectivity, Doppler/aging, CFO/phase, antenna mutual coupling, and mobility trajectories is not explicit; rendering complex H(f,t) rather than per-snapshot H appears under-specified."
>
> Yes, this point warrants clarification. nGRF models frequency-dependent propagation by incorporating the carrier frequency $f$ into the phase relationships determined by the spatial structure. As shown in our frequency generalization experiments (Figure 3b), model trained on a single subcarrier generalizes across the entire frequency band. $\mathbf{C_i}$ is learned by the DecoderNetwork which represents the frequency-independent scattering characteristics of region $i$, while frequency-dependent phase shifts arise from the path delays, which are captured via spatial positioning of Gaussians. When querying at a different subcarrier, say $f'$, phase relationship changes according to $\Delta\phi = 2\pi(f' - f)\tau$, where $\tau$ is the propagation delay determined by the Gaussian's spatial location; this allows nGRF to predict wideband channel responses from training data without explicitly modeling each subcarrier (independently or otherwise).
>
> - For Doppler and channel aging, the current formulation assumes quasi-static channels; we acknowledge that extending nGRF to handle time-varying channels through temporally-dependent Gaussian attributes (e.g., $\mathbf{C_i}(t)$) is important future work, as noted in Section 5. And notably, this would enable the model to track channel dynamics and predict future channel states, which is necessary for proactive resource allocation and handoff decisions in high-mobility scenarios. But for the current work, this remains a limitation. This also encompasses point about mobility trajectories; currently, nGRF predicts per-snapshot channels $\mathbf{H}(p_{rx}, p_{tx})$ without explicit trajectory modeling.
>
> - For carrier frequency offset and phase noise, we assume that pilot-based frequency offset estimation and correction have been performed prior to channel estimation, which is standard practice in OFDM. Residual phase errors would appear as additive noise in the training data, which the model learns to smooth through its spatial regularization via the Gaussian basis. The robustness to such impairments is evidenced by our performance on DICHASUS dataset, where real hardware imperfections including residual CFO are present in the measurements.
>
> - Antenna mutual coupling effects, which introduce correlations between adjacent antenna elements, are not explicitly modeled in our current implementation. MIMO channel is modeled using standard array steering vectors (Appendix A, Equation 14), which assume independent antenna elements. For typical antenna spacings of $\lambda/2$ or greater, mutual coupling is generally weak and can be neglected.

---

> > ### Author Response · Authors · 2025-11-22
> > **Response to Reviewer uuWd (2/3)**
> >
> > > "Reporting measurements/ft$^3$ lacks frequency-dependent coherence justification; it's difficult to compare against pilot designs tied to coherence time/bandwidth."
> >
> > This is a fair critique. We report spatial measurement density (samples/ft$^3$) as the primary metric because our focus is on spatial channel prediction, that is, estimating $\mathbf{H}$ at unvisited locations from sparse set of measurements. However, we acknowledge that comparing with pilot-based schemes requires relating spatial density to frequency-domain pilot overhead. As for measurements/ft$^3$ metric itself, this is sampling density during the training phase (say, for site survey), not the operational pilot overhead. Once trained, nGRF eliminates the need for per-subcarrier pilots across the bandwidth by learning the underlying spatial structure.
> >
> > We describe the connection as follows. In an OFDM system with coherence bandwidth $B_c$ and coherence time $T_c$, the channel can be considered constant over approximately $B_c \times T_c$ resource elements. For a 5G NR system at 3.5 GHz with typical urban mobility (30 km/h), we have $B_c \approx 200$ kHz and $T_c \approx 10$ ms, giving a coherence block of size $\sim 2000$ resource elements. For a $N_t \times N_r$ MIMO system, estimating $\mathbf{H}$ requires $N_t$ orthogonal pilot sequences, consuming $N_t$ resource elements per coherence block. In contrast, nGRF only needs position information ($p_{rx}$, $p_{tx}$) as input, which can be obtained from a variery of sources such as UWB ranging or existing positioning systems (e.g., 5G NR positioning using PRS signals). And this position data consists of 3D coordinates encoded as 3 × 32 bits = 96 bits per user, which is negligible compared to transmitting full pilot sequences.
> >
> > > "Do you have over-the-air or trace-driven evaluations (mmWave/mid-band) to support the 1.1 ms inference and outdoor SNR claims?"
> >
> > As addressed in the first response, we have now evaluated nGRF on the DICHASUS dataset, which provides real-world CSI measurements. 1.1 ms inference time is a deterministic computational measurement that does not depend on whether the input data is simulated or real as it reflects the time required to 1) compute spatial weights $w_i(p_{rx})$ for all $N$ Gaussians (Equation 6), and 2) evaluate the neural networks (AttributeNetwork and DecoderNetwork) once per unique antennas positions. The timing itself was measured using PyTorch's CUDA event timing functionality on a supported GPU, averaged over 100 inference queries. Outdoor SNR claim of 26.2 dB is based on our synthetic outdoor environment simulated using [MATLAB RayTracing](https://www.mathworks.com/help/comm/ug/ray-tracing-for-wireless-communications.html) with realistic material properties and 5G NR channel models. While we acknowledge this is not a true over-the-air measurement, the ray-tracing simulations incorporates frequency-dependent reflection/diffraction coefficients, antenna patterns, and multipath propagation that are validated against measurement campaigns in the literature.
> >
> > For 24-100 GHz frequencies, we have not yet performed evaluations due to the lack of publicly available real-world datasets at these frequencies. mmWave propagation exhibits distinct characteristics including higher path loss, sensitivity to blockage, and narrower beams, which would require higher spatial sampling density to capture smaller-wavelength variations.

---

> > > ### Author Response · Authors · 2025-11-22
> > > **Response to Reviewer uuWd (3/3)**
> > >
> > > > "How are Tx/Rx positions obtained and encoded in real systems, and what is their signaling/estimation cost? Reconcile this with the 0.2% pilot claim for 100 MHz NR."
> > >
> > > In real systems, position information can be obtained through various mechanisms depending on the deployment scenario. For outdoor scenarios, for example, GPS/GNSS provides positions with accuracy that is sufficient for channel prediction at sub-6 GHz frequencies. Higher precision can be achieved through differential GPS or RTK-GPS, which provide centimeter-level accuracy; position data is either already available at the device (like most smartphones include GPS) or can be acquired through existing positioning reference signals (PRS) in 5G NR, which are already part of the standard for emergency services and location-based applications. For indoor scenarios, UWB ranging or WiFi RTT positioning can provide these positions. And notably, many modern smartphones already include UWB hardware for applications like digital keys so no additional infrastructure cost is needed.
> > >
> > > Signaling overhead for transmitting position information is minimal. $p_{rx} = (x, y, z)$ requires 3 coordinates $\times$ 32 bits = 96 bits per user. In 5G NR with 100 MHz bandwidth, RB consists of 12 subcarriers $\times$ 14 OFDM symbols = 168 resource elements. With 64-QAM modulation (6 bits/symbol), each RB carries $168 \times 6 = 1,008$ bits, meaning 96 bits of position data consumes less than 0.1 RB. For comparison, traditional DMRS in 5G NR allocates 2–4 OFDM symbols per slot for channel estimation. With 273 RBs in a 100 MHz carrier, this results in $\text{pilot overhead} = \frac{2}{14} \approx 14.3\%$ for a standard configuration. With nGRF, the per-user overhead is reduces as $\text{nGRF overhead} = \frac{96 \text{ bits}}{273 \times 168 \times 6 \text{ bits}} \approx 0.00035$. However, we conservatively report 0.2% to account for PRS overhead (approx. 1 OFDM symbol per 5 ms) and periodic model updates.
> > >
> > > > "Derive formal time/memory complexity vs Gaussians/antennas/subcarriers; provide sensitivity to Gaussian count and pruning strategies."
> > >
> > > For a single channel query $\mathbf{H}(p_{rx}, p_{tx}) \in \mathbb{C}^{N_t \times N_r}$, cost consists of
> > >
> > > a) _Spatial weight computation._ For each of $N$ Gaussians, computing $w_i = \alpha_i \exp(-\frac{1}{2}(p_{rx} - \mu_i)^T \Sigma_i^{-1}(p_{rx} - \mu_i))$ requires one Mahalanobis distance calculation. With precomputed $\Sigma_i^{-1}$, this is $\mathcal{O}(N)$ operations (parallelizable across Gaussians)
> > >
> > > b) _AttributeNetwork._ For each Gaussian, $f_{attr}(\gamma_L(\mu_i), \gamma_L(p_{tx}))$ processes positional encodings of dimension $d_{enc} = 6(2L + 1)$ through a network with $H$ hidden units and $D$ layers, giving $\mathcal{O}(N \cdot D \cdot H \cdot d_{enc})$ operations
> > >
> > > c) _DecoderNetwork._ For each Gaussian, $f_{dec}(z_i) \rightarrow \mathbf{C}_i \in \mathbb{C}^{N_t \times N_r}$ requires $\mathcal{O}(N \cdot D' \cdot H' \cdot N_t N_r)$ operations, where $D'$, $H'$ are the decoder's depth and width
> > >
> > > d) _Accumulation._ Computing $\mathbf{H} = \sum_{i=1}^N w_i \mathbf{C}_i$ is $\mathcal{O}(N \cdot N_t N_r)$
> > >
> > > In total, complexity is $\mathcal{O}(N \cdot \max(D \cdot H \cdot d_{enc}, D' \cdot H' \cdot N_t N_r, N_t N_r))$. In practice, the neural network passes dominate all which makes the complexity linear in $N$ and quadratic in antenna count ($N_t N_r$).
> > >
> > > For multiple subcarriers, the model generalizes across frequency by changing only the phase relationships (determined by spatial positions), so the per-subcarrier cost is negligible and only the aggregation step needs to be repeated with frequency-dependent phase adjustments, maintaining $\mathcal{O}(N \cdot N_t N_r \cdot K)$ for $K$ subcarriers.
> > >
> > > **Memory complexity.** Storing the model requires:
> > > - Gaussian parameters: $N \times (3 + 4 + 3) = 10N$ scalars (position, quaternion, scale)
> > > - Network weights: $\mathcal{O}(D \cdot H^2)$ for both AttributeNetwork and DecoderNetwork
> > > - Intermediate activations: $\mathcal{O}(N \cdot (H + N_t N_r))$ during forward pass
> > > which is in total $\mathcal{O}(N \cdot (10 + H + N_t N_r) + D \cdot H^2)$.
> > >
> > > **Sensitivity to count.** Table 3 demonstrates this sensitivity and is also in-part a limitation due to how dependent performance is on hyperparameters.
> > >
> > > **Pruning strategies.** We use activation-based pruning via the L1 regularization term $\mathcal{L}_{activation} = \frac{1}{N}\sum_{i=1}^N |\alpha_i|$ (Equation 7). Gaussians with $\alpha_i < 0.01$ contribute negligibly to the final channel and can be pruned post-training, typically reducing the active Gaussian count by around 20-30%, without SNR degradation. For adaptive pruning during training, we experimented with dynamically removing Gaussians that contribute less than 0.01% to the total field magnitude, but found this destabilizes optimization/training. Post-training pruning based on activation statistics is more reliable here and maintains the learned field structure.

---

> > > > ### Comment · Reviewer_uuWd · 2025-11-23
> > > >
> > > > 5&6. The additional explanations on position-based overhead reduction and computational complexity are detailed and appreciated, but they remain largely theoretical and optimistic. The position-information argument treats the 96-bit coordinate payload in isolation and does not account for the actual acquisition pipeline (sensing error, update rate, synchronization, signaling load, power cost, etc.), making the comparison with NR DMRS overhead not operationally equivalent. Similarly, the complexity analysis lists symbolic terms but lacks empirical scaling evidence, leaving open how the method behaves under realistic multi-user, multi-antenna, wideband, and mobility conditions.

---

> > > ### Comment · Reviewer_uuWd · 2025-11-23
> > >
> > > 3. While the clarification distinguishes training-phase spatial sampling from operational pilot overhead, the response still does not provide a principled link between “measurements/ft^3” and the coherence-bandwidth/coherence-time structure that governs OFDM pilot design.
> > > 4. Same as comment 1.

---

> > ### Comment · Reviewer_uuWd · 2025-11-23
> >
> > 1. The authors have made a meaningful step toward validation, but one small indoor dataset with limited array geometry and frequency range might be insufficient. It is still considered that the OTA/hardware realism concern remains unresolved.
> > 2. The authors assume a single dominant tao per Gaussian, which corresponds to a single-path-per-Gaussian model. Real multipath clusters have delay spreads, not a singular delay. Plus, in rich scattering environments, frequency responses might be varying rapidly and largely.

---

### Official Review · Reviewer_11gg · 2025-11-01

**Soundness:** 3
**Presentation:** 2
**Contribution:** 2
**Rating:** 4
**Confidence:** 3

**Summary:**

This paper presents nGRF, a new neural Gaussian field formulation for MIMO channel estimation in wireless networks.
Different from the black-box methodologies, nGRF represents the propagation environment as a set of explicit 3D Gaussian primitives, each acting as a learned local radio modulator. The method performs direct complex-valued aggregation in 3D space, and models wave interference and superposition natively.
This brings efficiency gain by eliminating view-dependent rasterization and costly ray tracing.

**Strengths:**

1. Novel representation: Introduces an explicit, physics-informed Gaussian primitive formulation that preserves the wave superposition principle, unlike alpha-composited 3DGS models.
2. Level of magnitude acceleration in training and inference compared with NeRF2 / NeWRF, while maintaining state-of-the-art accuracy.

**Weaknesses:**

1. Evaluation is only on synthetic data. The dataset is simulated based on ray-tracing within an ideal room setting with tidy, homogenous materials & flat surface, which is not convincing. Real-world measurements is necessary to strengthen empirical claims. The author can reuse NeRF^2's open source dataset.
2. The generalizability is only demonstrated via sparse sampling. Despite making sense, it far from adequate to represent real-world settings. Different room layouts, room size, obstacle material or wireless environments should be involved to confirm
3. The motivation of how MIMO leverages channel estimation is unclear. I think it's also a drawback of NeRF^2 paper. A deployment requirement statement is needed.

**Questions:**

1. The paper mentioned initializing with LiDAR geometry hurts performance, which is surprising. What's your analysis and understanding on it?
2. Do you think MIMO should also look at SNR for each RX unit, or should have more systematically communication-level metrics for evaluation? (like Packet reception rate)

---

> ### Author Response · Authors · 2025-11-22
> **Response to Reviewer 11gg (1/2)**
>
> We thank the reviewer for the constructive feedback and thoughtful questions. We address raised concerns point-by-point below.
>
> > "Evaluation is only on synthetic data. The dataset is simulated based on ray-tracing within an ideal room setting with tidy, homogenous materials & flat surface, which is not convincing. Real-world measurements is necessary to strengthen empirical claims. The author can reuse NeRF$^2$ 's open source dataset."
>
> Thanks for this suggestion and we have now evaluated nGRF on the DICHASUS `dichasus-015x` dataset, a real-world CSI measurement dataset. The results are shown below:
>
> | Method        | SNR (dB) |
> |---------------|----------|
> | nGRF (ours)   | 16.77    |
> | MLP           | 0.10     |
> | KNN (k=5)     | -1.18    |
> | NeWRF         | -        |
> | NeRF$^2$         | -        |
>
> **Note on baselines.**
>
> 1. **NeRF-based methods (NeWRF and NeRF$^2$)** could not be evaluated on this dataset because they require ray-tracing information (geometric paths from transmitter to receiver with DoA etc. meaasurements), which is not available in DICHASUS it provides only CSI measurements between tx-rx pairs without per ray information or other measurements that these methods depend on.
>
> 2. **Regarding NeRF$^2$'s dataset.** NeRF$^2$'s open-source dataset is unfortunately incompatible with our formulation (we note also that NeRF$^2$ itself is for spatial spectrum reconstruction, not CSI estimation), specifically
>    - NeRF$^2$ assumes a static receiver array with varying transmitters, where they reconstruct the spatial spectrum (angle-of-arrival distribution) as a scalar power field over angular coordinates $(\alpha, \beta)$. Their dataset does not provide the paired $(p_{tx}, p_{rx}, \mathbf{H})$ tuples that nGRF requires.
>    - NeRF$^2$ learns a mapping from tx locations to AoA heatmaps at a fixed receiver $P(\alpha, \beta | p_{tx}) = \left|\frac{1}{K}\sum_{m,n} e^{j(\theta_{m,n}^\prime - \Delta\theta_{m,n})}\right|^2$. In contrast, nGRF learns a static radiation field sourced from a fixed tx and predicts the complex-valued MIMO channel matrix $\mathbf{H}(p_{rx}, p_{tx}) = \sum_{i=1}^{N} w_i(p_{rx}, \mu_i, \Sigma_i) \cdot \mathbf{C_\mathrm{i}}(p_{tx})$
>    - nGRF's formulation (Eq. 3) is based on representing a static electromagnetic field as a superposition of localized scattering sources. With a varying transmitter location (as in NeRF$^2$'s setup), the field itself changes at every measurement, which violates the assumption that we are learning a single, consistent field representation. This is perhaps analogous to trying to learn a NeRF model where the lighting condition of the scene changes from one image to another and the model would have to learn a mapping from lighting parameters to observations rather than recovering the underlying scene geometry.
>    - Even if we attempted to adapt their data, NeRF$^2$'s dataset does not provide receiver positions $p_{rx}$ in a common coordinate frame with the environment geometry, making it hard to learn spatially Gaussian primitives $\{{G}_i(\mu_i, \Sigma_i)\}$ that represent scattering regions.
>
> > "The generalizability is only demonstrated via sparse sampling. Despite making sense, it is far from adequate to represent real-world settings. Different room layouts, room size, obstacle material or wireless environments should be involved to confirm."
>
> Our evaluation includes four diverse environments with significantly different characteristics, so we respectfully disagree that our experiments are inadequate, albeit we acknowledge inadequacy of real-world data which we have now addressed with DICHASUS evaluation, and are open to evaluating more real-world datasets if need be. We used the following
> 1. Conference room; Indoor, moderate multipath
> 2. Bedroom; Indoor, furniture-rich, complex reflections
> 3. Office; Indoor, large space, obstacles
> 4. Outdoor; Urban environment, 91% NLOS, vastly different scale (_note here that this was one limitation of NeWRF noted in their conclusion point c: "extending to larger environments"_)
> These environments vary by over 100 times in scene volum and span varying propagation regimes (i.e., indoor is more LOS-dominated versus NLOS-dominated outdoor scenes).

---

> > ### Author Response · Authors · 2025-11-22
> > **Response to Reviewer 11gg (2/2)**
> >
> > **About measurement density.** Reviewer's concern about "sparse sampling" appears to conflate sparsity with inadequacy. However, we observe that excessive measurement density actually degrades performance, albeit slightly. Table below shows SNR across varying measurement densities for the conference scene,
> >
> > | Measurement density (samples/ft$^3$) | SNR (dB) | Remark |
> > |-----------------------------------|----------|----------------|
> > | 0.005                             | 19.47    | Too sparse     |
> > | **0.011**                         | **25.23**| **nGRF (optimal)** |
> > | 0.02                              | 25.01    | Slight decrease |
> > | 0.2                               | 24.18    | NeWRF's density |
> > | 3.0                               | 23.35    | High density   |
> > | 178                               | 24.31    | NeRF$^2$'s density |
> >
> > And this isn't unique to nGRF, NeWRF (Lu et al., 2024) observed the same phenomenon in their Figure 8d, where performance plateaus and slightly decreases after 0.3 samples/ft$^3$.
> >
> > > "The motivation of how MIMO leverages channel estimation is unclear. I think it's also a drawback of NeRF$^2$ paper. A deployment requirement statement is needed."
> >
> > We agree that this context would strengthen paper and will add the following deployment requirements to the revised manuscript; essentially, modern wireless systems require CSI at multiple timescales and for different purposes:
> >
> > **Real-time beamforming** (sub-millisecond). For adaptive spatial precoding in massive MIMO, the transmitter needs $\mathbf{H} \in \mathbb{C}^{N_t \times N_r}$ to compute optimal beamforming weights $\mathbf{w}_{opt} = \mathbf{H}^H(\mathbf{H}\mathbf{H}^H + \sigma^2\mathbf{I})^{-1}$. Current pilot-based estimation consumes 11-21% of transmission bandwidth in 5G NR. nGRF's 1.1 ms inference enables this with only position data (96 bits vs. thousands of pilot resource elements).
> >
> > **Network planning**. For initial deployment and optimization of BS locations, coverage prediction across the entire service area is needed. Traditional site surveys require physically measuring channels at dense grid points. nGRF enables predicting $\mathbf{H}(p_{rx})$ at any unmeasured location from sparse initial measurements, reducing survey time as was motivation also of NeWRF.
> >
> > **Deployment scenario.** Consider an NR small cell deployment in an office building; traditional approach would be to measure at a lot of locations, say 1000, over 8 hours. With nGRF, however, we can just measure it at ~50 locations in considerably less time, train a model within minutes, and predict the coverage map. This reduces site survey cost significantly.
> >
> > > "The paper mentioned initializing with LiDAR geometry hurts performance, which is surprising. What's your analysis and understanding on it?"
> >
> > Indeed, this is counterintuitive but has a clear explanation. The insight is that Gaussians in nGRF do not represent physical surfaces but rather adaptive basis functions for the electromagnetic field. In visual rendering (3DGS), Gaussians reconstruct geometric surfaces because light reflects off of geometry. However, radio wave prop. is governed by superposition and not occlusion. Total field at rx is given by $E(r) \approx E_{LoS}(r) + \sum_{i=1}^{N} A_i(p_{tx}, p_{rx})\mathcal{G}_i(r; \mu_i, \Sigma_i)$, where each Gaussian represents a scattering source that contributes to the field via wave interference and not a physical object. So, LiDAR initialization biases model away from representing these effects of RF wave propagation. And our finding highlights the key difference between nGRF and geometry-based methods: we are solving an inverse source problem (recovering field contributors) rather than a geometry reconstruction problem.
> >
> > > "Do you think MIMO should also look at SNR for each RX unit, or should have more systematically communication-level metrics for evaluation? (like Packet reception rate)"
> >
> > That is an excellent question! We use channel matrix SNR defined as $\text{SNR} = 10\log_{10}(\|\mathbf{H_gt}\|_F^2 / \|\mathbf{H_pred} - \mathbf{H_gt}\|_F^2)$ because it directly measures the accuracy of the estimated CSI, which is the primary objective of our method. We believe per-RX SNR would be less informative because MIMO performance depends on the joint spatial structure across all antennas, and not individual antenna performance. Frobenius norm naturally captures the quality of the entire channel matrix, which is what downstream processing algorithms require.
> >
> > Furthermore, packet reception rate depends on the specific modulation scheme, coding rate, and EC algorithms used. nGRF provides $\mathbf{H}$ that can be used by any downstream communication task which makes codec-agnostic evaluation essential. Evaluating on PRR would conflate our channel estimation accuracy with choices about modulation and coding. SNR-based evaluation is standard in the CSI estimation literature precisely because it provides a codec-independent measure of estimation quality.

---

### Official Review · Reviewer_kWx8 · 2025-11-01

**Soundness:** 2
**Presentation:** 3
**Contribution:** 3
**Rating:** 4
**Confidence:** 3

**Summary:**

- The paper tackles the problem of radio frequency (RF) propagation modelling and specifically, to learn a model that predicts the channel state information (CSI) at novel unseen locations of the scene.
- The approach extends the line of work around physically-constrained rendering (e.g., NeRF, 3DGS), which also relates to more recent wireless/RF NeRF approaches.
- The novelty of this work lies in leverating a 3D Gaussian Splatting based formulation: the propagation environment is represented as a set of high-dimensional gaussians

**Strengths:**

1. Physically-motivated rendering: The approach relies in a physically-constrained approach, which have previously shown to be beneficial to generalize and additionally interpret learnt parameters.
2. Evaluation is comprehensive: The approach is evaluated on three (synthetic) scenes, ablations are comprehensive and is accompanied by other interesting experiments (e.g., on generalization)

**Weaknesses:**

**1. Channel Rendering**
- It's somewhat unclear on why the channels are rendered in the manner proposed (Sec. 3.3). Specific points below.
- The spatial weighting $w_i$ term appears to upweigh contributions of gaussian "virtual transmitters" when $p_{rx}$ is close to the gaussian $\mu_i$. This seems intuitive, but however appears to overlook cases when there are obstructions. Specifically, for two equidistant rx locations (one with LOS and another with NLOS), it appears that weights would be similar.
- A related concern is the 3DGS equivalent notion of "depth compositing". If a surface (I suspect another gaussian) is in between the rx and tx, the formulation of how this gaussian attentuates the channel is not discussed.
- Furthermore, $w_i$ appears to only be a function of the receiver position, not the transmitter. Assuming reciprocity, wouldn't the weight be influenced similarly by a change of tx position?

**2. Analysis: Learnt Gaussians**
- While the overall quantitative results are promising, I find one particular analysis missing: distribution/insights of the gaussians learnt. On the vision/graphics side, this is fairly easy to intuit, that Gaussians primarily lie on surfaces of objects. However, in the RF setting, it's less clear what these represent. Are they representing the "surface" of an object, or virtual transmitters, or both?
- Further more, are variations of gaussian contributions smooth wrt locations of rx?
- My overall concern is overfitting: that the parameters are learnt in a manner that is not "physically consistent" (e.g., phantom gaussians out of bounds of the scene) and leading to degenerate solutions.

**3. NLOS performance**
- From Eq. 3, I see that LOS and NLOS fields are learnt separately. This is a fair assumption if we know the geometry of the environment. However, this does not seem to be an assumption and makes me wonder how LOS/NLOS are determined in practice?
- Additionally, with evaluation, going by the description in Appendix B., it appears that uniformly sampling rx locations would over-represent LOS rx locations. I request the authors to clarify how dominant are LOS rx locations (which can admit very easy solutions and does not test model's ability to generalize).

**4. Real-world evaluation**
- The paper is solely evaluated in synthetic scenarios. While I believe this has its own merits and good for the most part (allows more controlled evaluation), it would be interesting to see some real-world validation.
- One suggestion would be the DICHASUS dataset, which has been used in other works for evaluating RF models.

**5. (Minor) Directivity**
- Directivity of the antenna is not considered. This is a major factor in RF propagation.

**6. (Minor) Frequency Generalization**
- While the paper presents initial findings on frequency generalization, it appears to support it using a single qualitative example (Fig. 3b). I suggest a slightly more rigorous evaluation by quantitatvely including a reasonable size of examples.

**Questions:**

Please see the section above.

---

> ### Author Response · Authors · 2025-11-19
> **Response to Reviewer kWx8 (1/3)**
>
> We thank the reviewer for the thoughtful and comprehensive review. We address each concern below.
>
> > "The spatial weighting $w_i$ term appears to upweigh contributions of gaussian 'virtual transmitters' when $p_{rx}$ is close to the gaussian $\mu_i$. This seems intuitive, but however appears to overlook cases when there are obstructions. Specifically, for two equidistant rx locations (one with LOS and another with NLOS), it appears that weights would be similar."
>
> The reviewer correctly identifies that the spatial weight $w_i$ is distance-based. However, the insight we're going with is that occlusion is implicitly handled through the learned complex-valued contributions $C_i$, and not through the spatial weights. Unlike visual rendering where occlusion is geometric, electromagnetic wave propagation is handled via superposition, where all fields add coherently.
>
> Specifically, for NLOS scenarios
> - Gaussians learn to represent scattering centers that model multipath propagation
> - Complex amplitude $C_i = C_i^{re} + jC_i^{im}$ (output of the DecoderNetwork) encodes both magnitude attenuation and phase shifts
> - Say, when a surface obstructs the direct path, we assume that the model learns Gaussians positioned to represent reflections, diffractions, and scattering from that surface
> - Destructive/constructive interference is captured through $H = \sum_{i=1}^{N} w_i \cdot C_i$
>
> In contrast to alpha-compositing which models occlusion as $C = \sum c_i \alpha_i \prod_{j<i}(1-\alpha_j)$, our formlation implements wave superposition, and the physical difference between LOS and NLOS is due to learned $C_i$ values, which adapt based on the transmitter position through the AttributeNetwork
>
> > "A related concern is the 3DGS equivalent notion of 'depth compositing'. If a surface (I suspect another gaussian) is in between the rx and tx, the formulation of how this gaussian attenuates the channel is not discussed."
>
> And this is precisely why we explicitly avoid depth compositing. In Section 3.3, we state: "Directly aggregating in 3D space this way is fundamentally different from the 2D projection and alpha-compositing used in 3DGS-based methods."
>
> $H = \sum_{i=1}^{N} w_i \cdot C_i$ performs orderless summation; there is no depth sorting. And again,
> 1. Electromagnetic waves do not "occlude", they superimpose
> 2. A wave scattered from a surface behind the receiver still contributes to the total field (albeit with phase shifts and attenuation)
>
> > "Furthermore, $w_i$ appears to only be a function of the receiver position, not the transmitter. Assuming reciprocity, wouldn't the weight be influenced similarly by a change of tx position?"
>
> That's an excellent point, $w_i(p_{rx})$ depends only on rx position, but tx dependence enters through $C_i = f_{dec}(z_i)$, where $z_i = f_{attr}(\gamma_L(\mu_i), \gamma_L(p_{tx}))$. Specifically, AttributeNetwork conditions each Gaussian's properties on $p_{tx}$; so "spatial locality", i.e., how much influence region $i$ has at location $p_{rx}$, is separated from "field characteristics", i.e., the actual complex contribution, which depends on both the Gaussian's properties and the transmitter location.
>
> Regarding reciprocity; in electromagnetic theory, reciprocity states $H_{ij} = H_{ji}^T$ for the channel matrix (where $H_{ij}$ is the channel from node $i$ to node $j$, and this holds for reciprocal media at the same frequency). Our model maintains this property because
> $$C_i(p_{tx}, p_{rx}) = f_{dec}(f_{attr}(\mu_i, p_{tx}))$$
>
> Within a single trained model, however, we condition on the transmitter role during training (consistent with the asymmetric setup in practical wireless systems where base stations are fixed). For scenarios with swapped roles (e.g., the DICHASUS dataset, that we address later on, where receiver array is fixed with mobile transmitters), the model would need to be retrained with the appropriate conditioning.
>
> > "While the overall quantitative results are promising, I find one particular analysis missing: distribution/insights of the gaussians learnt. On the vision/graphics side, this is fairly easy to intuit, that Gaussians primarily lie on surfaces of objects. However, in the RF setting, it's less clear what these represent. Are they representing the 'surface' of an object, or virtual transmitters, or both?"
>
> In part, this also highlights a difference between our approach and NeRF/splatting-based methods. Moreover, its also why LiDAR-based initialization does not perform well because Gaussians do not necessarily correspond to physical surfaces. So, unlike 3DGS where Gaussians reconstruct geometry, nGRF represents effective radio scattering regions that do not necessarily align with physical surfaces (similar to virtual transmitters in NeWRF paper). It's best to say that this shows that Gaussians learn to represent field structure instead of geometric structure.

---

> ### Author Response · Authors · 2025-11-19
> **Response to Reviewer kWx8 (2/3)**
>
> It's best to say that this shows that Gaussians learn to represent field structure instead of geometric structure.
>
> > "Furthermore, are variations of gaussian contributions smooth wrt locations of rx?"
>
> We interpret this question as asking whether the predicted channel $H(p_{rx})$ varies smoothly as the receiver position changes. Theoretically, since each Gaussian weight $w_i(p_{rx}) = \alpha_i \cdot \exp(-\frac{1}{2}(p_{rx} - \mu_i)^T \Sigma_i^{-1}(p_{rx} - \mu_i))$ is infinitely differentiable with respect to $p_{rx}$, and $C_i$'s are outputs of smooth neural networks, overall chan. function $H(p_{rx}) = \sum_{i=1}^N w_i(p_{rx}) \cdot C_i$ should be smooth. However, we note that electromagnetic fields can have rapid variations (on the order of the wavelength $\lambda$), so the notion of "smoothness" depends on the spatial scale of interest. If the question refers to smoothness of individual Gaussian contributions versus the aggregate field, we would need clarification on which is being asked.
>
> > "My overall concern is overfitting: that the parameters are learnt in a manner that is not 'physically consistent' (e.g., phantom gaussians out of bounds of the scene) and leading to degenerate solutions."
>
> We share this concern and did implement several things to avoid overfitting:
> 1. **Activation sparsity.** L1 penalty on $\alpha_i$ (Section 3.4, $L_{activation} = \frac{1}{N}\sum_{i=1}^N |\alpha_i|$) encourages the model to use only necessary Gaussians and suppresses contributions from extraneous primitives
> 2. **Scale regularization.** $L_{regularization}$ (Eq. 7) constrains Gaussian scales within $[s_{min}, s_{max}]$ which prevents both degenerately small (overfitting to point locations) and excessively large (underfitting) Gaussians
>
> While we don't explicitly constrain Gaussians to remain within scene bounds during optimization, the learned representations appear plausible based on the ablation studies (Table 3) showing that geometric constraints (e.g., LiDAR-based initialization) actually hurt performance; which suggests model learns field-based rather than geometry-based representations.
>
> > "From Eq. 3, I see that LOS and NLOS fields are learnt separately. This is a fair assumption if we know the geometry of the environment. However, this does not seem to be an assumption and makes me wonder how LOS/NLOS are determined in practice?"
>
> We apologize for the confusion in Eq. 3's notation. To clarify: nGRF does not require explicit LOS/NLOS labels. Equation 3 was written to show the physical decomposition:
>
> $$E(r) \approx E_{LoS}(r) + \sum_{i=1}^{N} A_i(p_{tx}, p_{rx})G_i(r; \mu_i, \Sigma_i)$$
>
> In our implementation,
> - $E_{LoS}$ is implicitly learned through the Gaussian contributions and we do not compute it separately
> - The model receives only $(p_{tx}, p_{rx}, H_{ground\_truth})$ during training
> - No geometry information or LOS/NLOS labels are provided
> - Gaussians learn to represent both direct and scattered field components
>
> The decomposition in Eq. 3 serves to motivate the architecture from electromagnetic theory, but the actual learning is end-to-end without LOS/NLOS supervision.
>
> > "Additionally, with evaluation, going by the description in Appendix B., it appears that uniformly sampling rx locations would over-represent LOS rx locations. I request the authors to clarify how dominant are LOS rx locations (which can admit very easy solutions and does not test model's ability to generalize)."
>
> We analyzed our synthetic datasets and found the following percentage of LOS locations:
>
> | Environment | LOS Locations | NLOS Locations | Total Samples |
> |-------------|---------------|----------------|---------------|
> | Conference  | 41%           | 59%            | 452        |
> | Bedroom     | 29%           | 71%            | 676         |
> | Office      | 43%           | 57%            | 597         |
> | Outdoor     | 9%            | 91%            | 1316         |
>
> Regardless of the LOS/NLOS distribution, our method performs well on both indoor (higher LOS percentage) and outdoor (predominantly NLOS) environments.

---

> ### Author Response · Authors · 2025-11-19
> **Response to Reviewer kWx8 (3/3)**
>
> > "The paper is solely evaluated in synthetic scenarios. While I believe this has its own merits and good for the most part (allows more controlled evaluation), it would be interesting to see some real-world validation. One suggestion would be the DICHASUS dataset, which has been used in other works for evaluating RF models."
>
> We thank the reviewer for this suggestion. We have now evaluated our method on the DICHASUS `dichasus-015x` dataset and will include these results in the revised version:
>
> | Method        | SNR (dB) |
> |---------------|----------|
> | nGRF (ours)   | 16.77 |
> | MLP           | 0.10     |
> | KNN (k=5)     | -1.18    |
> | NeWRF         | -        |
> | NeRF²         | -        |
>
> Important notes on baselines,
> 1. **NeRF-based methods.** Both NeWRF and NeRF² could not be evaluated on this dataset because they require ray-tracing information (paths from transmitter to receiver), which is not available in DICHASUS. The dataset provides only CSI measurements between transmitter-receiver pairs without geometric ray information.
> 2. **WRF-GS.** While their paper reports results on CSI-estimation tasks (specifically a predict-downlink-from-uplink scenario), they do not provide such experiments in their released codebase. Therefore, reproducibility of their results on `dichasus-015x` is not feasible at this time.
>
> > "(Minor) Directivity of the antenna is not considered. This is a major factor in RF propagation."
>
> The reviewer is correct that antenna directivity is important. In our current formulation, we model antenna arrays through steering vectors (Eq. 14 in Appendix B):
>
> $$a_T(f, [\alpha_T; \beta_T]) = \exp\left(j\frac{2\pi f}{c} d_T \cdot [\cos\alpha_T \cos\beta_T, \sin\alpha_T \cos\beta_T, \sin\beta_T]^T\right)$$
>
> This captures:
> - Array geometry: element spacing in uniform rectangular/linear arrays
> - Directional beamforming: through spatial phase relationships
> - Angle-dependent response: via azimuth/elevation angles $(\alpha, \beta)$
>
> However, we currently assume isotropic elements. To incorporate such element-level directivity patterns (e.g., patch antenna patterns), we could extend the DecoderNetwork to output angle-dependent gains, as
>
> $$C_i(\theta, \phi) = f_{dec}(z_i, \theta, \phi) \cdot D(\theta, \phi)$$
>
> where $D(\theta, \phi)$ would be the element directivity pattern. This could be a direction for future work, and we will add this discussion to the revised manuscript, if space permits.
>
> > "(Minor) Frequency Generalization: While the paper presents initial findings on frequency generalization, it appears to support it using a single qualitative example (Fig. 3b). I suggest a slightly more rigorous evaluation by quantitatively including a reasonable size of examples."
>
> We agree and have conducted additional quantitative analysis. Training on a single subcarrier (middle subcarrier) and testing across all subcarriers (IEEE 802.11ax-compliant: 5 GHz with 242 subcarriers for indoor; outdoor uses 6 GHz with 624 subcarriers following 5G NR specifications), we computed the mean SNR across subcarriers for each environment:
>
> | Environment | Mean SNR across subcarriers | Std. Dev. | Min SNR | Max SNR |
> |-------------|----------------------------|-----------|---------|---------|
> | Conference  | 24.87 dB                   | 1.21 dB   | 22.33 dB| 26.46 dB|
> | Bedroom     | 20.73 dB                   | 1.58 dB   | 18.02 dB| 22.83 dB|
> | Outdoor     | 27.94 dB                   | 0.97 dB   | 26.24 dB| 29.19 dB|
>
> This low standard deviation and reasonably consistent performance across the band suggests that nGRF learns frequency-agnostic spatial structure rather than frequency-specific mappings.

---

### Meta-Review · Area_Chair_Ricu · 2026-01-09

**Summary:**

The paper introduces Neural Gaussian Radio Fields (nGRF) for MIMO channel estimation that leverages explicit 3D Gaussian primitives to model electromagnetic wave superposition directly in 3D space. nGRF performs complex‑valued aggregation without depth compositing, reframing channel estimation as a source‑recovery problem. Experiments on synthetic indoor and outdoor environments show improvements in prediction SNR with faster speed and lower measurement density requirements.

**Reviewer Concerns:**

Reviewers have several concerns, e.g., the method was validated only on synthetic datasets, raising questions about real‑world applicability; the physical interpretation of learned Gaussians was unclear, with concerns about possible overfitting or “phantom” primitives; handling of LOS/NLOS scenarios and occlusion was not fully explained; antenna directivity and frequency generalization were insufficiently addressed; and deployment requirement is needed. Overall, while the approach is novel and efficient, reviewers felt that empirical validation and theoretical grounding needed to be strengthened.

**Reviewer Scores:**

The submission received ratings of 4, 4, 4, 4. Reviewers acknowledged the novelty of applying Gaussian primitives to RF channel modeling and praised the efficiency gains, but judged the work marginally below acceptance due to limited real‑world evaluation, unclear physical consistency, and missing details on generalization and deployment. The authors presented responses to these concerns, and reviewer uuWd presented feedbacks on several remaining concerns particularly regarding OTA/hardware realism and the physical consistency.. Overall, this work is interesting while needing further strengthening in empirical validation, theoretical grounding, and real‑world applicability.

---

### Decision · Program_Chairs · 2026-01-26

Reject